# Perceptography unveils the causal contribution of inferior temporal cortex to visual perception

Elia Shahbazi [1] ✉, Timothy Ma[2], Martin Pernuš [3], Walter Scheirer[4] & Arash Afraz [1]

Neurons in the inferotemporal (IT) cortex respond selectively to complex visual features, implying their role in object perception. However, perception is subjective and cannot be read out from neural responses; thus, bridging the causal gap between neural activity and perception demands independent characterization of perception. Historically, though, the complexity of the perceptual alterations induced by artificial stimulation of IT cortex has rendered them impossible to quantify. To address this old problem, we tasked male macaque monkeys to detect and report optical impulses delivered to their IT cortex. Combining machine learning with high-throughput behavioral optogenetics, we generated complex and highly specific images that were hard for the animal to distinguish from the state of being cortically stimulated. These images, named "perceptograms" for the first time, reveal and depict the contents of the complex hallucinatory percepts induced by local neural perturbation in IT cortex. Furthermore, we found that the nature and magnitude of these hallucinations highly depend on concurrent visual input, stimulation location, and intensity. Objective characterization of stimulation-induced perceptual events opens the door to developing a mechanistic theory of visual perception. Further, it enables us to make better visual prosthetic devices and gain a greater understanding of visual hallucinations in mental disorders.

Artificial stimulation of neurons in high-level visual cortical areas induces hallucinatory percepts[1–4], the experience of complex visuals in the absence of corresponding retinal stimulation. Scientific characterization of these visual percepts poses a serious challenge due to their complex and subjective nature, yet it has inspired a multigenerational effort in systems neuroscience as it bridges the causal gap between patterns of neuronal activity in the brain and elements of visual perception[5–7]. From a translational point of view, understanding the causal underpinnings of visual hallucinations induced by local brain stimulation is necessary to develop prosthetic devices that restore vision by direct brain stimulation[8,9]. This knowledge also

shapes the building blocks for understanding visual hallucinations in mental disorders and altered states of consciousness[10–12].

In this study, we created a machine learning structure and used it in combination with high-throughput behavioral optogenetics in macaque monkeys in order to, for the first time, produce pictorial descriptions of the perceptual events induced by brain stimulation in the high-level visual cortex. These pictorial descriptions, called *perceptograms*, provide unbiased and parametric yet rich accounts of the visual perceptual events following optogenetic activation of ~1 mm³ neural subpopulations in the inferior temporal (IT) cortex. The basic idea behind our quest was simple: guided by the animals' behavior, is it

[1]National Institutes of Health (NIH), Bethesda, MD, USA. [2]Center for Neural Science, New York University, New York, NY, USA. [3]Laboratory for Machine Intelligence (LMI), University of Ljubljana, Ljubljana, Slovenia. [4]Department of Computer Science and Engineering, University of Notre Dame, Notre Dame, IN, USA. ✉e-mail: elia.shahbazi@nih.gov

possible to evolve specific image perturbations that resemble the sense of being stimulated in a given cortical locus in the absence of physical stimulation?

We performed viral injections in the central IT cortex of two macaque monkeys (*Macaca mulatta*) in order to express the excitatory opsin C1V1 under the CaMKIIa promoter in a ~5 × 5 mm area of the cortex. We then implanted arrays of LEDs (Opto-Array, BlackRock Neurotech) on the virally transduced cortical area as well as the corresponding position in the opposite hemisphere where no viral injection was performed. The Opto-Array allows safe, rapidly reversible, and high-throughput optical stimulation of ~1 mm³ subregions of the targeted cortex, although it doesn't allow neural recordings. Technical details about the Opto-Array and relevant surgical protocols can be found in our earlier reports[4,13,14].

The two monkeys were trained to detect and report a brief optogenetic stimulation impulse delivered to their IT cortex while fixating on a 1-second sequence of images created by a generative adversarial neural network (GAN) (Fig. 1a). It has been previously shown that monkeys can easily learn this simple task[15,16], which remained the sole task expected from the animals throughout the study. Our earlier results suggest that the animals perform this task (in the IT cortex) using the visual events induced by cortical stimulation[4]. The animals initiated each trial by holding fixation on a central target

for 500 ms. Then, a natural-looking GAN-generated image was shown for 400 ms (seed image) on a gray background. The image subtended 8°×8° of visual angle, and the animals were required to hold fixation at its center throughout the trial. Next and in all trials, the seed image would turn into a randomly perturbed version of itself for 150 ms, then turn back into the original image and stay changeless for 450 ms. In half of the trials, randomly selected, at the time of image perturbation, an LED was activated on the animals' IT cortex for 150 ms, typically at 3 mW photometric power. After the sequence of images (seed-perturbed-seed), the screen was cleared, and two response targets appeared on the vertical midline (white, 0.4° diameter, 5° above and below the center). The animals then made a saccade to one of the two targets in order to indicate if the trial included a brain stimulation impulse (chance level 50%). The response targets then disappeared, and the animals received a liquid reward for correct responses and 3.5 s of timeout for incorrect responses. Trials with broken fixation or latency greater than 3 s for making a response were aborted and discarded. These trials were injected into the future stream of trials in a pseudorandom order.

As reported in our earlier work, the animals learned to perform the cortical perturbation detection (CPD) task quickly while fixating on static images (without any image perturbation), and they were not able to detect cortical illumination over the intact cortical area where no

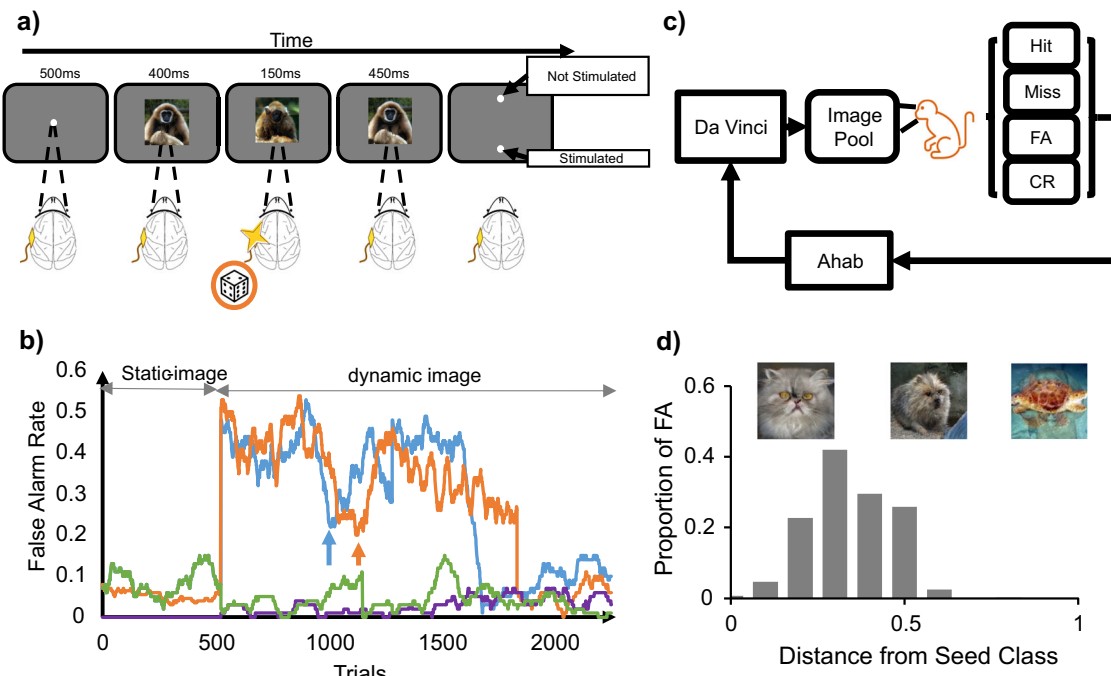

**Fig. 1 | Perceptography paradigm and pipeline. a** Cortical perturbation detection (CPD) task. After fixation, a short movie consisting of a 400 ms presentation of a seed image followed by 150 ms of a perturbed image, and then 450 ms of the original seed image was played. In 50% of trials at random, an ~1 mm³ locus in the IT cortex was optogenetically stimulated for 150 ms at the same time as the perturbed image presentation. The animals were rewarded for correctly identifying if trials contained brain stimulation or not. **b** The first training days with dynamic stimuli. The abscissa shows trials, and the ordinate represents the false alarm rate. The first 500 trials represent the animals' performance before switching to the dynamic image sequence (initial training for the task). The rest of the plot shows the FA rate after the 150 ms image perturbation was first introduced to the training regime. Both animals first took the image alterations as "stimulated trials" at very high rates but learned within a few hundred trials to ignore most of the image perturbations and veridically detect the cortical stimulation. The small vertical arrows indicate the end of the first training day for each monkey. Blue: Monkey Sp, Orange: Monkey Ph. (Miss rate: purple: Sp, and green: Ph) **c** Perceptography pipeline. The illustrator engine, DaVinci, generated a pool of randomly perturbed images. The optimizer

engine, Ahab, analyzed the monkeys' performance in the CPD task to extract the image features that increased the likelihood of behavioral false alarms. Ahab sent the optimized parameters to DaVinci to generate new pseudo-random image perturbations. These Ahab-optimized images were heavily diluted with random DaVinci images and injected back into the image pool for the next cycle of perceptography. **d** Proportion of behavioral false alarms as a function of the magnitude of image perturbation. The abscissa shows the normalized feature distance (based on the BigGAN interpolation factor) of each randomly perturbed image from its seed image. The ordinate represents the behavioral FA rate for the first pool of DaVinci images. Each bin contains 440-470 non-stimulated trials. The feature distance represents the ratio of BigGAN non-class features included in each image generated by the engine (see methods). A value of 0 indicates that all features belong to the seed class, and one indicates that all features are sourced from other classes. Images above the chart are examples of the visual change corresponding to the distances of 0, 0.5, and 1 from the seed class (class 283 of BigGAN), respectively, from left to right.

viral injection was performed[4]. We noticed that the animals' performance in detecting cortical stimulation varied by the choice of fixated images. Moreover, and to our surprise, we learned that presenting images on the screen generally helps the animals in detecting IT stimulation, with the lowest performance observed when the animals viewed a blank screen at the time of brain stimulation[4,17]. This suggests that the animals rely on the perceptual distortions induced in the fixated images to detect cortical stimulation. Given the lower dynamic range of performance for the blank screen as well as the technical complications of mutating blank images, we avoided perceptography with a blank seed for this initial study. We aim to explore this complicated yet interesting matter systematically in our following studies.

Following training with static images, image sequences (seed-perturbed-seed) were introduced for further training, and the animal task remained the same: detection of cortical stimulation. On the first day, when the dynamic image perturbation in the middle of the trial was introduced, both animals mistakenly mixed the image perturbations with cortical stimulation, and as a result, their false alarm (FA) rate dramatically increased from 8 and 5.2 percent to 39.2 and 37.6 percent, respectively for monkeys Sp and Ph. This sudden increase in FA rate cannot be the result of a general increase in task difficulty because the Miss rate remained unchanged (see Fig. 1b). This suggests that optogenetic stimulation of the IT cortex induces a "visual" perturbation that can be mixed up with an image perturbation on the screen. Note that the FAs are the trials in which no cortical stimulation was delivered, yet the animal reported the trial as "stimulated." Also, note that an FA is considered a behavioral mistake and is never rewarded, nor are the Miss trials, those on which a stimulated trial is reported as non-stimulated. Nevertheless, at this stage, we cannot strictly reject the possibility that the animals generalize a non-visual sense of transience induced by brain stimulation to the fixated images. Within a single day, both animals learned to discriminate IT stimulation from the image perturbations on the screen and performed the task with high performance at 90.2 and 89 percent correct and only 8.3 and 6.2 FA rates (respectively for Sp and Ph). This remarkable observation is documented in Fig. 1b.

## Results

### The state of brain stimulation can be mimicked by images

After the training phase and once stable high-performance levels (above 80% for both animals) were achieved, the animals entered the first phase of behavioral data collection. While the monkeys performed the simple CPD task for tens of thousands of trials, under the hood, two learning systems controlled the experiment with the goal of evolving specific image perturbations that increase the chance of behavioral false alarms. We refer to these two systems as DaVinci and Ahab (see Fig. 1c). DaVinci is our image illustrator engine, a structure powered by BigGAN trained on the ImageNet dataset[18,19]. DaVinci was tasked with creating multiple random image mutations for each seed image (see Methods). Ahab is our feature-vector optimizer (see Methods) tasked with tracking the animals' behavioral responses to DaVinci's random image perturbations. Ahab learned from the animals' behavioral mistakes and gave feedback to DaVinci to produce image perturbations that would increase the FA rate. An increase in the FA rate (trials without stimulation reported as stimulated) could result from a general increase in task difficulty, which would also increase the Miss rate (trials with stimulation reported as non-stimulated). To avoid this, Ahab was set to aim at specifically increasing the FA rate without changing the Miss rate (see Methods).

The image evolution process started with 5-6 image seeds; for each seed, DaVinci created 400–1000 randomly perturbed images. Each of these image perturbations was presented to the animal at least five times in the course of multiple days (a total of 10–30K behavioral trials). While image perturbations are done randomly over a nearly infinite feature-vector space (see Methods), the amplitude of these perturbations varied: small perturbations randomly but subtly change the image, while large perturbations induce random yet massive pictorial alterations. Figure 1d plots the behavioral FA rate as a function of image perturbation magnitude.

The non-monotonic relationship observed here indicates that high FA rates cannot be achieved simply by increasing the magnitude of image perturbations. Instead, since the animals actively search for a particular image distortion, the one induced by brain stimulation, they are more likely to be tricked by the image perturbations that match the magnitude of the stimulation-induced perceptual event. We found this result encouraging as it shows that the behavioral false alarm rate in the CPD task can be systematically manipulated by altering the image. The distribution of behavioral false alarms over image alterations of various sizes reflects the magnitude of the perceptual perturbations induced by cortical illumination for the stimulation intensity used in this experiment (3 mW).

### Artificial intelligence learns from the brain how to trick it

Next, Ahab scored each image perturbation and selected the ones that induced a higher FA rate without increasing the Miss rate (see Methods). Ahab guided DaVinci to create an image family for each surviving image, including the original image and 2–6 mutated children. These image families were then presented to the animals in the context of the next round of behavioral testing, and the images that were scored high by Ahab received the chance to mutate again and make their own children. This process was repeated until at least one of the image families passed the threshold of 60% FA over at least 12 presentations. This typically took five iterations of the entire process, involving 1–5 K Ahab-optimized image presentations. The image that scored highest within a winning family was named a *perceptogram*, as viewing it was hard for the animal to distinguish from the perceptual state induced by brain stimulation. The entire process was accordingly coined: *Perceptography*. Throughout the course of each round of perceptography, a single LED of the Opto-Array was selected and used. The intensity of the LED was adjusted at each new cortical position in order to keep the behavioral output under ceiling performance.

The process of Perceptography, if successful, would increase the FA rate across generations of images. This could untrain the animals over the course of time because we only reward objectively correct choices. To avoid untraining the animals by this procedure, we heavily diluted Ahab-optimized image families with non-optimized DaVinci images as the evolution progressed (50–80 percent non-optimized). While the optimized images were heavily diluted by DaVinci images, the animal's FA rate kept increasing specifically for those images as the evolution progressed. Figure 2a shows the monkeys' FA rate as a function of session number for DaVinci and Ahab optimized image families. As shown in the figure, the FA rate remained at a constant level of 2.8–4.1% and 4.1–6.1% (respectively for Sp and Ph) for DaVinci images, but Ahab optimized image families induced more FAs increasingly as the process unfolded. Figure 2b, c shows the evolution process for a typical perceptogram, starting from a large variety of image perturbations and converging to a specific one.

### Robustness of the results

Is it possible that some image perturbations survive through the pipeline by chance without being meaningful to the animals? We bootstrapped the data, but instead of letting the animals determine the distribution of FAs for each trial, we distributed them randomly. Figure 3a shows the results. If the false alarms were randomly distributed across image presentations, the best image family would have a cumulative false alarm rate significantly lower than the image families selected by the perceptography process. More interestingly, data shows that the contents of these behaviorally selected images are related. In fact, as Fig. 3b shows, the images selected independently by the animals' behavior across families share increasingly more features

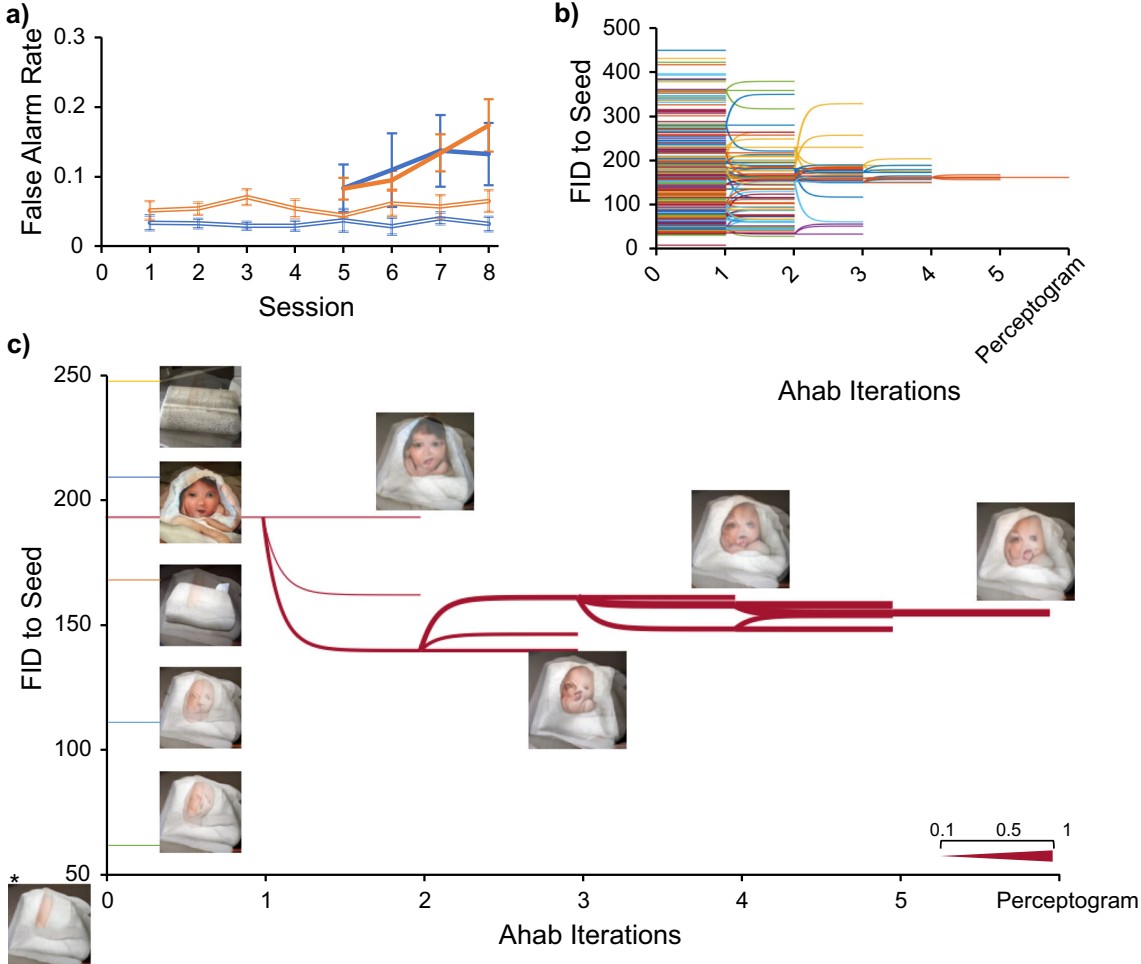

**Fig. 2 | Evolution of perceptograms. a** The false alarm rate for non-optimized (Davinci, double lines) and optimized (Ahab, solid lines) perturbed images. The abscissa indicates the progress of perceptography across sessions. The ordinate shows the FA rate. Blue: Sp, Orange: Ph. Ahab optimized images induced a significantly higher false alarm rate (df = 74 and 76, $p$ = 0.025 and 0.002 for Sp and Ph, respectively, Welch's $t$-test). Error bars indicate ±1 standard error of mean. **b** Evolution dendrogram. Each colored line represents a single image family. To survive the iterations of Ahab optimization, image families had to maintain a cumulative false alarm rate of over 50%. The ordinate shows the Fréchet Inception Distance (FID) between each perturbed image and its corresponding seed image. The abscissa shows iterations of the perceptography procedure. **c** Example of a perceptogram image family tree. The abscissa and ordinate are the same as in subplot (**b**). The legend on the bottom right shows how the thickness of each branch corresponds to its false alarm rate. Five examples of image mutations from the initial DaVinci pool are shown together with the winning image family tree. The asterisk indicates the seed image.

as the process unfolds. These analyses show that using the image evolution process presented here, it is unlikely to get an image tagged as a perceptogram just by chance. They also show that the image evolution process is not a random stray trajectory; instead, it is systematically guided by the animals' choices converging on specific answers. Despite these statistical encouragements and in order to fully cross-validate these findings with a fresh set of data, we performed the entire perceptography procedure on the same image seed, cortical position, and stimulation intensity once again for each animal. Figure 3c shows how two independent perceptography procedures converged on similar answers. These procedures, each lasting ~17 work days, were performed 24 and 10 days apart from each other in monkeys Sp and Ph, suggesting that the perceptual effects of repeated optogenetic stimulation in a given cortical position remain stable at least over the course of about one month.

A design feature of the experiments reported here is that we use the same image perturbations in both stimulated and non-stimulated trials. This balancing feature is crucial in order to take away all the potential image cues and forces the animals to perform the task only by detecting the cortical stimulation impulse. This feature, however, introduces a measurement uncertainty to the process. As a result of

stimulus balancing, all stimulated trials include two perturbation components: one is on the screen, and the other comes from the brain stimulation. The screen component is not informative and varies at every trial, so the monkey is incentivized to ignore it and detect only the cortical component. Now, when viewing a perceptogram in a non-stimulated trial, the animal matches the perceptogram to the net perceptual effect of cortical stimulation (constant across stimulated trials) plus a baseline random non-informative component (variable across trials). While this introduces an inherent uncertainty in the procedure, in that the measurement process affects the measure of interest, since the image perturbations are mostly small and random, the average perturbed image is not expected to drift far from the original seed image. From the point of view of an incentivized observer, most image perturbations are expected to be perceived as irrelevant, except the ones that warp the seed image in the same direction as induced by the brain stimulation. If true, this should increase the chance of reporting the trial as stimulated in both stimulated and non-stimulated conditions. Figure 4a shows that the hit rate is higher than baseline when the cortex is stimulated while looking at perceptograms. Although, given the high baseline hit rate, the reward that the monkey gains at stimulated trials (grand average 7.9% and 6.5% for Sp

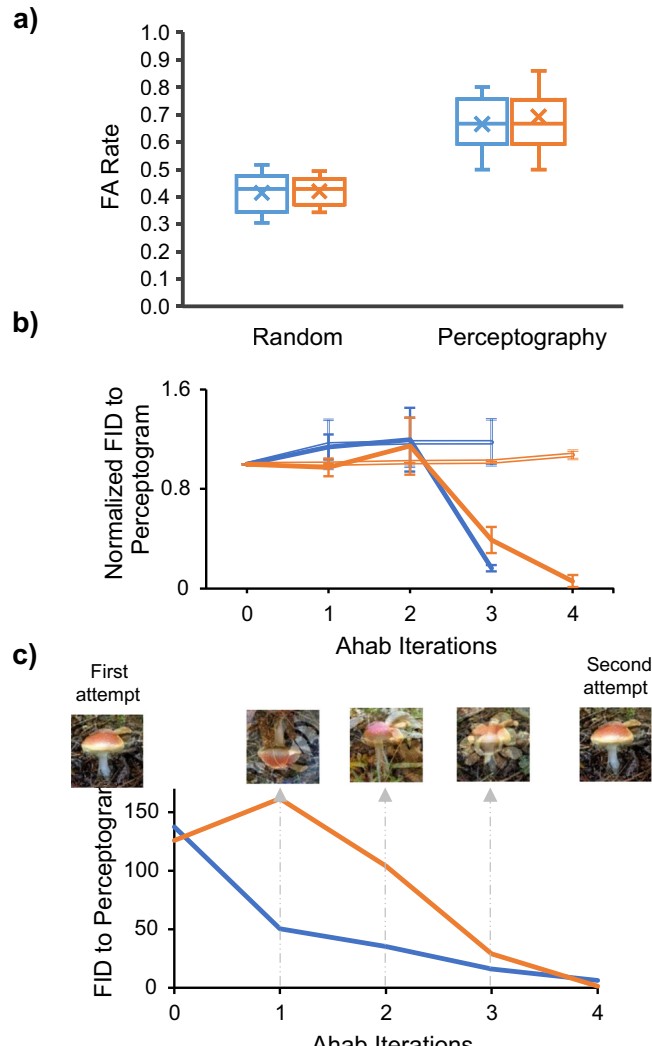

**Fig. 3 | Evolution trajectory of perceptograms; random or guided? a** High FA rates cannot be achieved by random selection of image families across iterations. Left: the distribution of maximum false alarm rates achievable in bootstrapped data where FA scores are randomly assigned to images at each iteration of perceptography. Perceptograms, images that evolved guided by the animals' behavior, had significantly higher FA rates compared to the best images produced by the bootstrapping procedure (df = 13 and 17 for Sp and Ph, respectively, $p < 0.0001$ for both, Welch's $t$-test). Right: the distribution of perceptogram false alarm rates. Blue: Sp, Orange: Ph. Error bars indicate the minimum and maximum rates. **b** Convergence to similar images across Ahab iterations. The abscissa represents Ahab's iterations of optimization. The ordinate shows the FID feature distance. The solid lines represent the FID distance of the final perceptogram from images in each optimization iteration, excluding the perceptogram family. Independently optimized images get more similar to each other, and the final perceptogram as the process unfolds. Double lines represent the same for the bootstrapped data where family survival is randomly chosen. As the image pool was optimized by Ahab, the distance (FID) between the optimized pool and the to-be-discovered perceptogram decreased. Note that the images were not selected for similarity but based on the behavioral FA rate they evoke (Blue: Sp, Orange: Ph). Error bars indicate ±1 standard error of mean. **c** Independent evolution of similar perceptograms. Two independent rounds of perceptography were performed for each monkey (Blue: Sp, Orange: Ph). The axes are similar to the subplot (**b**). The line plot shows the FID distance of the optimized images of the second round of perceptography with the perceptogram obtained from the first round.

and Ph) is far less than the reward loss at non-stimulated trials when perceptograms are presented (grand average 60.0% and 64.3% for Sp and Ph). Moreover, it seems that the animals psychophysically rely on contrasting stimulation with the solid seed images presented before

and after the stimulation more than the perturbed image itself. In an experiment, we showed image perturbations of one seed image (150 ms) temporally sandwiched between images of another seed. This was done for two seed images in each monkey. The FA rate dramatically decreased in all cases, indicating that the perceptual effect of stimulation is perceived and matched by the animals mainly in temporal contrast to the seed image. Specifically, the false alarm rate dropped to 0% and 2% (out of 50 presentations) in Sp and Ph, respectively. (Hit rate 98% and 100% in Sp and Ph, respectively).

The appearance of the perturbed image induces apparent motion, is it possible that the perceptography procedure selects images with high motion energy because they interact more with the CPD task due to motion-based masking? We analyzed the motion energy for Perceptoram sequences as well as a hundred randomly chosen sequences from the initial DaVinci pool. The results showed no significant difference in motion energy between the two groups of image sequences (t(279) = 0.11, $p = 0.46$). Also, note that motion-based masking would induce a general increase in task difficulty and cannot explain the higher hit rate when perceptograms were presented in the stimulated trials.

When matching an image sequence to a perceptual state induced by cortical stimulation, how should the response latency of IT be considered? IT neurons have a response latency of ~70-80 ms. It is not clear what part of the IT response causally contributes to perception, thus, any time adjustment would be based on arbitrary assumptions. While the potential effect of stimulation timing needs to be systematically studied using short impulses and various time lags, for the current study, which in our view is a proof of concept, we applied no time correction but used a relatively long (150 ms) stimulation impulse to overlap with the IT response. We argued that the potential subtle temporal difference in the timing of the events would be difficult for the monkeys to notice, especially in the stimulation-absent trials that are the source of the FAs. Moreover, if the animals noticed a potential time lag effect, they would have used it in order to get more rewards and not be tricked by the perceptograms. Nevertheless, optimization of the image and stimulation temporal profile might improve the FA rate and remains an interesting area to explore.

### Effects of stimulation intensity
Figure 4b shows examples of perceptograms obtained from the two animals. As an independent sanity check, we hypothesized that if a perceptogram truly reflects the perceptual changes induced by cortical stimulation, the magnitude of image perturbation in the winning perceptograms should increase if we increase the cortical stimulation intensity. To test this, we performed independent perceptography procedures on similar cortical positions, each at two different cortical illumination powers (1 and 2 mW for Sp, 1 and 3 mW for Ph). Figure 5a, b demonstrate that the amount of feature warping in the winning perceptograms was remarkably higher when higher cortical illumination was applied. Examples of perceptograms at each level of cortical illumination are shown in Fig. 5c. Consistently, the baseline miss rate of both animals was slightly but significantly lower in the high illumination condition, as shown in Fig. 5d.

### Effect of cortical position
While comparing the perceptograms coming from different LED channels, we noticed that the anterior channels induced more holistic changes in the image. While perceptograms express significant pixel deviations from their seed images all along the posterior-anterior axis, the quality of these changes varies systematically. Inspecting the examples presented in Fig. 6, it is apparent that stimulation in the posterior channels of the array distorts the perceived image by adding unrelated visual features to the contents of perception. However, the anterior channels induce perceptual changes that are identity-preserving. These subjective evaluations can be tested by state-of-

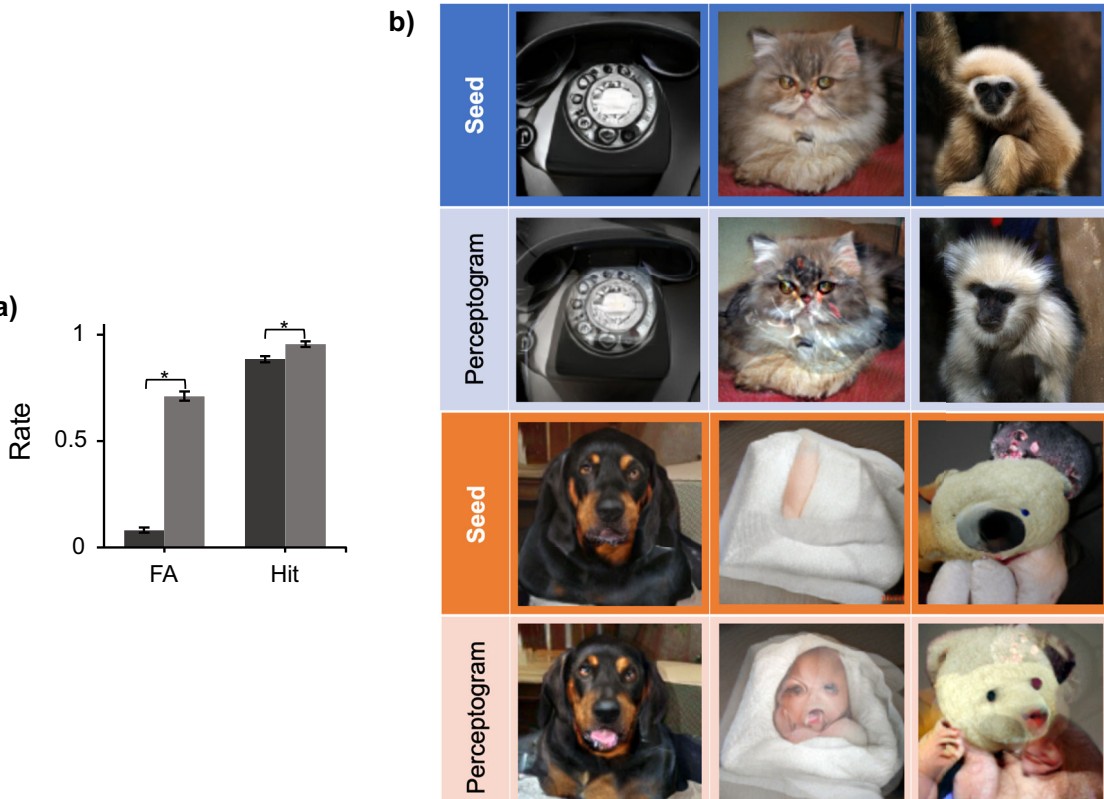

**Fig. 4 | The effect on hit rate and some examples of perceptograms.**
**a** Perceptograms increase the hits as well as FAs. The false alarm rate evoked by the perceptograms (light gray) is significantly higher than that of the non-optimized DaVinci image pool (dark gray) (df = 30, p < 0.001). The hit rate is also significantly higher in perceptograms even though the effect is smaller due to a ceiling effect (df = 30, p < 0.001). Error bars indicate ±1 standard error of mean. **b** Examples. Three examples are shown from each monkey (Blue shades: Sp, Orange shades: Ph); in each block, the top row indicates the seed images, and the bottom row shows their corresponding perceptograms.

the-art object classification tools. Analysis of images shows that perceptograms of the anterior channels of the array tend to retain general features of the seed image as shown by the high confidence in image classification and a low FID distance to the seed (measuring Frechet Inception Distance score, calculates feature vector distance between generated and real images, or any two sets of generated images, Fig. 6b–d). In contrast, perceptograms of the posterior LEDs express the opposite effect, where additional features are introduced, thus lowering the confidence in image classification and increasing the FID to the seed. Consistent with numerous studies of IT cortex that show a tendency for neural responses to more holistic features along the posterior-anterior axis of the cortex[7,20–24], this finding supports the causality of the relationship. Alternatively, given that the array spans only 5 mm of the cortex (25-30% of the length of the posterior-anterior axis), it is possible to attribute the spatial systematicity in the structure of perceptograms simply to the heterogeneity of cortical function at the mm scale.

## Potential neural underpinnings

Overall, the development of each perceptogram cost ~30–50 K behavioral trials, collected in the course of 14–20 work days. We performed a total of 32 complete rounds of perceptography over seven cortical locations (3 and 4 for monkeys Sp and Ph) and 15 seed images. These results provide pictorial evidence of the visual perceptual hallucinations induced by stimulation of the high-level visual cortex. Examples of a few perceptograms are shown in Fig. 4b. These results show that it is possible to behaviorally exchange the state of local brain stimulation in IT cortex with the state of viewing an image. The similarity of the two states is close enough to make the animals choose to tag ~70% of the non-stimulated perceptogram trials as stimulated, even at the cost of losing reward. While an "ideal perceptogram" is expected to induce a 100% FA rate, the ones found in this study (mean FA rate 70.2%, Median = 71%, StD = 12) are surprisingly close, given the very low baseline FA rates. The residual from 100% can be due to the imperfection of our image generation engine and/or potential effects of stimulation that are impossible to mimic on a 2D screen (e.g., 3D hallucinations, nonvisual feelings, etc.). Such effects, even if existing, must be very subtle in amplitude because the animals are incentivized to use any clue to receive a reward.

What is the relationship between perceptograms and the preferred stimuli of their driving neurons? IT cortex is known for its strong object selectivity at the single cell[25,26] as well as ~1 mm³ tissue scale[21,27,28]. While the current OptoArray technology doesn't allow neural recording, rendering us blind with respect to the object selectivity profile of the stimulated neurons, it is reasonable to assume heterogeneity of selectivity at the spatial scale perturbed by a single LED[4,21] in that the perturbed neural population conserves visual preference for a part of the shape space. Is perceptography simply another way to measure the stimulus preference of the stimulated neurons? Not necessarily. "Preferred stimuli" of neurons reveal how the visual signal is encoded in IT cortex, and Perceptograms show how the signal gets decoded from IT by the rest of the brain. These two do not necessarily match, and the relationship between the two can vary under different decoding frameworks. In some cases of sensory processing, neurons are tightly tuned to specific physical stimuli. Activation of such a neuron induces the appearance of its related sensory stimulus in perception. Such a direct one-to-one hypothetical relationship between the preferred stimulus of a sensory neuron and the percept it arises is known as the

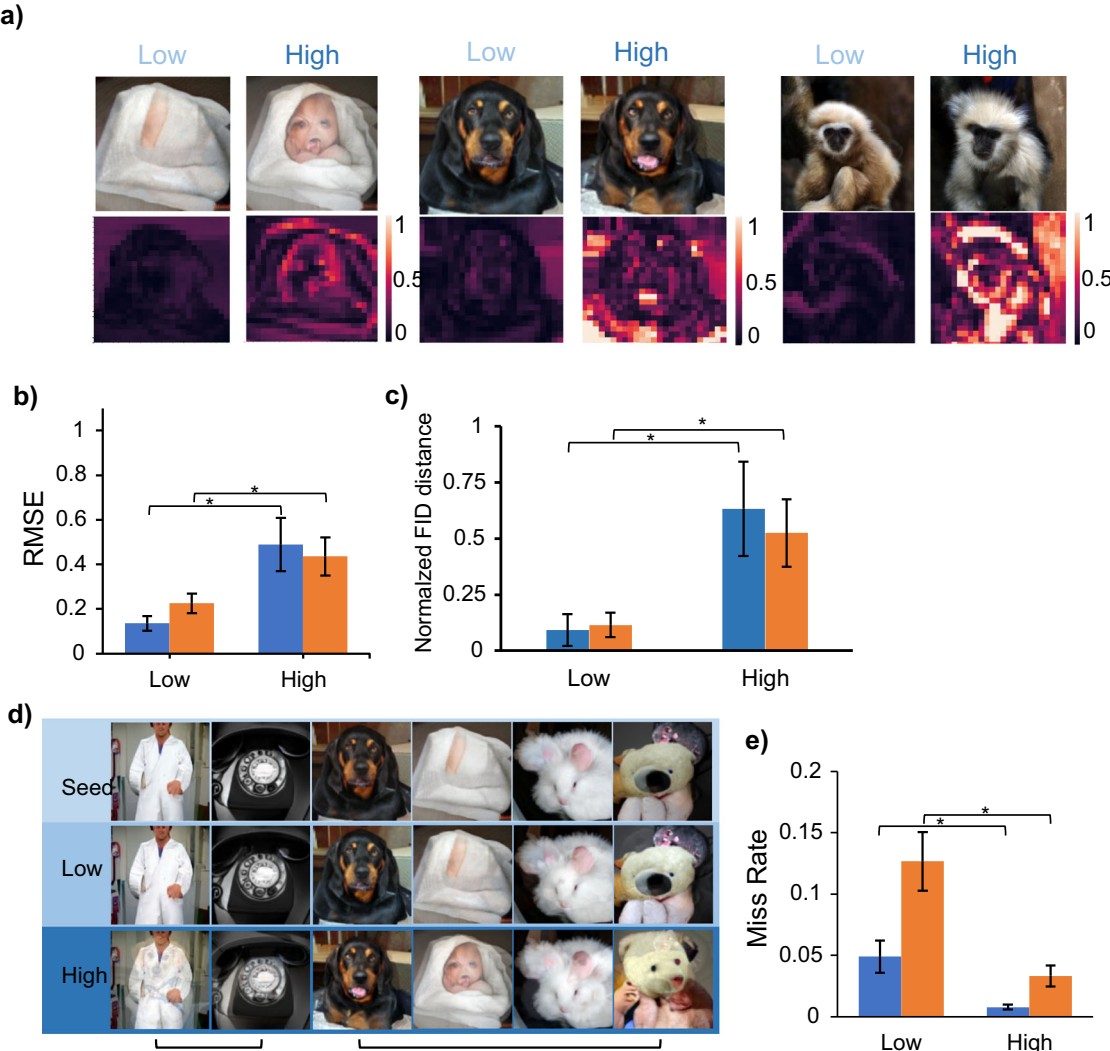

**Fig. 5 | Effects of stimulation intensity. a** Examples of heat maps depicting the image changes between the perceptograms and their corresponding seeds. In each block of heatmaps (left and middle block Ph, right block Sp), the left column includes the perceptograms obtained from low illumination perceptography. In contrast, the right column depicts the high illumination perceptograms in the same LED channel and monkey. **b** More intense cortical illumination warps the resulting perceptograms further away from their seed images. In both animals, the perceptograms obtained with higher intensity of stimulation had significantly higher distances from their image seeds compared to the perceptograms resulting from low-intensity cortical illumination (Root Mean Squared Error (RMSE): df = 8 and 13, *p* = 0.025 and 0.026 for Sp and Ph respectively, Welch's *t*-test). Blue: Sp, Orange: Ph. RMSE is the square root of the average of the squared differences between corresponding pixels in the two images. Error bars indicate ±1 standard error of mean. **c** Same effect with a different measure. The abscissa is the same as in 5.b. The ordinate shows normalized feature-vector distance (df = 8 and 13, *p* = 0.001 and 0.0002 for Sp and Ph, respectively, Welch's *t*-test). Blue: Sp, Orange: Ph.).
**d** Examples of perceptograms obtained with low and high stimulation intensities. Top: seed images. Middle: perceptograms obtained with low cortical illumination. Bottom: perceptograms obtained with higher illumination power. The brackets under the subplot indicate the perceptograms obtained from the same channel. **e** The effect on the behavioral miss rate. Increasing the illumination intensity of the LEDs significantly decreased the behavioral miss rate in both monkeys (df = 13 and 8, *p* = 0.026 and 0.023 for Sp and Ph, respectively, Welch's *t*-test). Blue: Sp, Orange: Ph. Error bars indicate ±1 standard error of mean.

labeled line hypothesis. Alternatively, more complex decoding frameworks might govern the relationship between neuron's stimulus preference and their causal impact on perception. For instance, a medium wavelength cone on the retina responds mostly to the green light, but its activation does not necessarily induce perception of the color green. The perceived color, in this case, depends on the activation ratio of other cone types as well as the position of the activated cone in the retinal cone mosaic[29,30]. Now, is the decoding schema of IT cortex a labeled line or a coarse code[31] like the case of color? Our results are not consistent with the labeled line framework. Assuming the labeled line hypothesis, one expects that stimulation of a given site in IT cortex induces perception of the preferred features of the targeted neurons independent of what is presented to the eyes. If this is the case, examination of perceptograms is expected to reveal common

visual elements in the perceptograms obtained from the same channel. The results, though, show a completely different picture. Figure 5c depicts examples of perceptograms obtained from one cortical position in each of the two monkeys along with their corresponding seed images (more examples are provided in the supplementary materials, Supplementary Fig S1). The first property that is apparent in perceptograms is that their structure strongly depends on the seed image. Perceptograms that come from stimulation of a single point in IT cortex are typically very different from each other, lacking at least an obvious explicit common visual element. An analysis of the perceptograms using Yolo, a real-time object detection system[32] revealed that 82% (StD = 21%) of the class-labels in the perceptograms are shared with their corresponding seed images. The analysis also showed that the added class-labels (compared to the seed) of the perceptograms

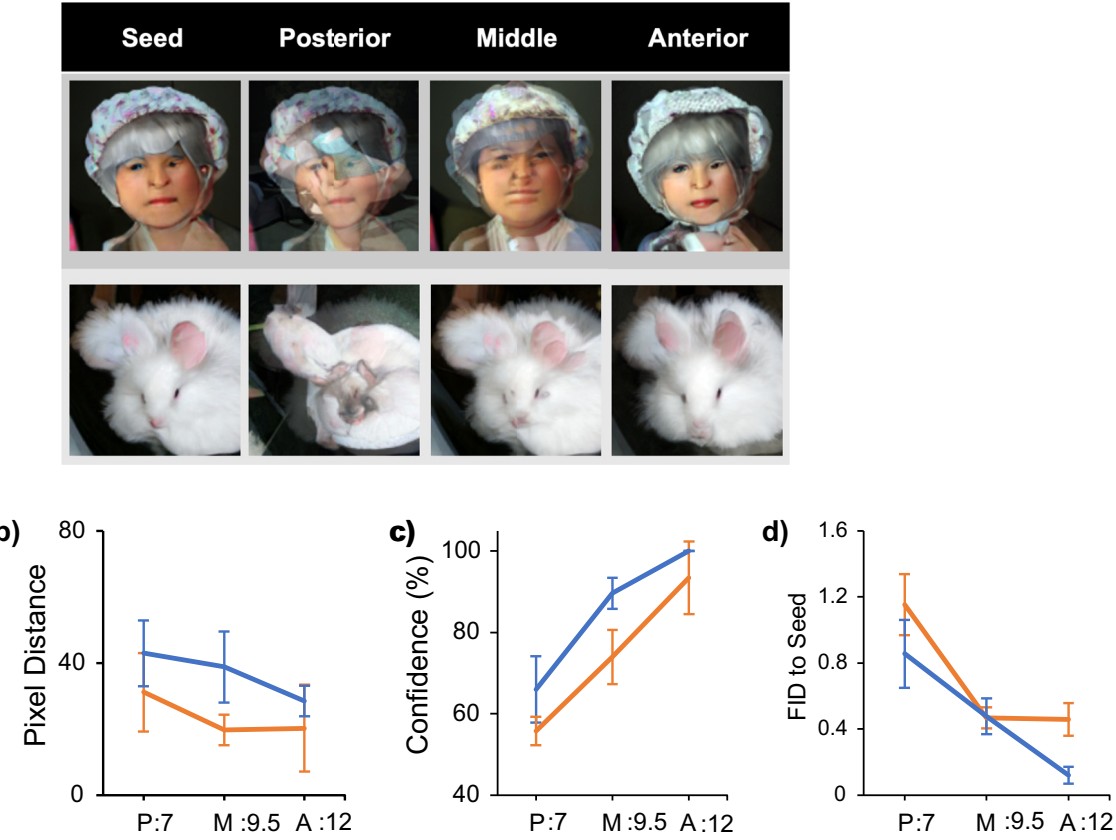

**Fig. 6 | Effects of cortical position on perceptograms. a** Examples of perceptograms obtained along the posterior-anterior axis of the central IT cortex. **b** Pixel distance of seed to the perceptogram. The abscissa represents the stimulation position relative to the interaural line along the posterior-anterior anatomical axis of IT cortex. The ordinate shows the pixel distance of the perceptograms resulting from each AP position from one seed image. Blue: Sp, Orange: Ph. While all perceptograms show pixel distance from their seed images, the effect does not change across cortical AP positions on this measure, and the line graph is statistically flat (df = 8 and 11, *p* = 0.202 and 0.197 for Sp and Ph, respectively, ANOVA). **c** Classification confidence of a Yolo (real-time object detection system) fed by perceptograms obtained from different cortical positions on the posterior-anterior axis. The abscissa is the same as in (**b**), and the ordinate shows classification confidence. Blue: Sp, Orange: Ph. Classification confidence significantly increases for the perceptograms obtained from anterior cortical positions (df = 8 and 11, *p* = 0.026 and 0.005 for Sp and Ph, respectively, ANOVA). **d** FID distance from the seed. The abscissa is the same as in **b, c**. The ordinate shows the FID of the perceptograms obtained from each LED to its seed image. Blue: Sp, Orange: Ph. FID, normalized to mean, significantly differs across cortical positions (df = 8 and 11, *p* = 0.009 and 0.011 for Sp and Ph, respectively, ANOVA). Error bars indicate ±1 standard error of mean for all subplots.

acquired from a single channel have only a little in common with each other (0% and 7% in Sp and Ph, respectively), which is not different (*t*-test, *p* > 0.4 for both animals) from the overlap between the added class-labels obtained from different channels (0% and 10% in Sp and Ph respectively). This suggests that the pattern of neural activity in the cortex, which varies by the seed image, strongly influences the outcome of local stimulation in IT cortex. This is consistent with the recent findings about the vast activity landscape of IT neurons[33] and the idea that the activity of a neural unit is interpreted by the rest of the brain only in the context of the state of other similar neural units[7]. These findings strongly encourage recording the neural activity together with perceptography, a point that is further dissected in the *conclusion.*

Another point emerging from the examination of the set of perceptograms produced in our experiments (Supplementary Fig. S1) is that most of the perceptograms show image changes that are off the manifold of natural objects. However, a few seem suspiciously natural; for example, a dog seed image (Fig. 4b bottom block) has turned into exactly the same dog sticking out its tongue, or a monkey (Fig. 4b top block) has turned into a very similar monkey with long light-colored hair and the head turned a few degrees. Consistent with this observation, a scoring algorithm based on Yolo[32], scored 15% (3 out of 20) of

the perceptograms as "natural images" (defined as less than 10% change in the main label confidence compared to the seed without introducing any new label with confidence more than 20%). This shows that perturbing the neural activity in ~1 mm³ of IT cortex forces the neural state off its natural manifold on most occasions; however, in some cases, the pattern of activity induced by the external stimulus is so that the same neural perturbation creates a naturally meaningful change. Determining when a perturbation lands on the natural manifold of neural activity is a critical step for breaking the code that maps the neuronal activity to perception[7].

## Discussion

Constructing a mechanistic theory of visual perception requires establishment of causal homeomorphism between the neural state, a system measured in units of spikes per second, and the perceptual state, a system measured in psychophysical units[6,7]. Making the bridge between the two requires parametric characterization of both. However, simultaneous measurement of both in large primate brains poses a serious technical challenge. In an ideal setup and in order to close the gap between perception and neural activity, recording of the brain state is needed to measure the neural effects of brain stimulation and the selectivity profile of the targeted cells. We appreciate the

importance of neural recordings; nevertheless, we argued that neural recording is an existing concept that can be added later to the toolset, but accurate measurement of subjective percepts is a conceptual challenge, thus, it should be the first problem to tackle. Therefore in this study, given the existing limitations in optogenetics technology, we decided to focus on characterizing the perceptual events induced by neural stimulation as it has been a historical and methodological bottleneck. This modified challenge had two faces of its own; one required reliable high-throughput stimulation in a large brain, and the other demanded custom-tailored artificial intelligence in order to develop effective perceptograms. As for the first one, we chose optogenetics over traditional electrical stimulation as it provides more accurate and more interpretable stimulation capacity given that it does not target axons of passage[34,35], and it is less invasive by being a surface implant[13]. Furthermore, electrical stimulation is not reliable for the high number of stimulation trials required here[36]. The second face of the challenge demanded not only searching a very large image space but also mimicking the effect of stimulation well enough to deceive the animals against reward. Ahab controlled the search function (see methods), and DaVinci mastered mimicking images by combining two GAN-generated images to achieve an accurate reconstruction of images outside its original training set (see methods).

Facing this two-faced challenge, perceptography provides pictures that are behaviorally exchangeable, to a good degree, with the state of being cortically stimulated. Given the parametric nature of these pictures, we can now provide objective and quantitative evidence of the nature and quality of stimulation-driven visual perceptual effects. This allows measurement of the causal contribution of a given neural group into the perceptual space[37]. Characterization of this causal contribution, once combined with descriptions of neural sensory responses[33], establishes the missing link between neural activity and perception. This can be done in the context of quantitative modeling of the decoding theory that links the two. While existing fully theoretical models of visual hallucinations yield encouragingly comparable results to our observations[38,39], further research is needed in order to complete the picture. Completion of these steps will provide access to the building blocks of a potential unifying mechanistic theory of perception and consequently provide a deeper understanding of visual hallucinations in mental disorders. It also allows the development of better visual prosthetic devices. Visual prosthetic devices traditionally target the primary visual cortex. This forces the prosthetic system to recreate any visual scene by "phosphene" elements, the result of local stimulation in the primary visual cortex. However, it is challenging to restore a rich and complex visual experience only by shapeless phosphene elements[40,41]. The current manuscript documents the high-level visual effects induced by stimulation of IT cortex. Understanding these high-level visual distortions allows us to control them and use them, potentially besides phosphenes, as building elements for recreation of the visual experience.

Altogether, given that the amount of work left to be done in this important area is practically beyond the working bandwidth of a single lab, we find this adventure incomplete yet mature enough to be shared with the scientific community. We hope this work sparks interest in those interested in underlying mechanisms of visual perception and encourages technique developers to invest in platforms that allow easy, high-throughput simultaneous recording and stimulation of the cortex in large brains.

## Methods

We conducted experiments and gathered data involving two adult male Rhesus monkeys (Macaca mulatta), named Sp and Ph. The details of the complimentary surgical and anesthesia procedures, along with postoperative care and methods of implantation, were thoroughly documented in a prior publication[14]. All procedures were conducted in accordance with and approved by the National Institute of Mental Health Animal Use and Care Committee guidelines.

### The optical array

We injected AAV5-CaMKIIa-C1V1(t/t)-EYFP (nominal titer: $8 \times 1012$ particles/ml) in the cortex using a custom-made injection array consisting of four 31-gauge needles arranged in a $2 \times 2$ mm square[42]. We tiled the central IT cortex four times with sixteen evenly spaced injection loci, resulting in a ~$6 \times 6$ mm viral transduction area. At each injection site, 10 μl ($10 \, mm^3$) of the virus was injected at the rate of 0.5 μl/min, totaling an injection volume of 160 uL ($160 \, mm^3$). After each injection, a 10-min wait period was introduced before array removal to allow for virus diffusion into the cortical tissue to ensure uniform viral expression.

We later implanted OptoArrays (Blackrock Neurotech - 530 nm wavelength) on the virally transduced area as well as the same anatomical region in the opposite hemisphere not injected with the virus. The 3D models of the animals' brains and skulls were reconstructed with the FLoRIN method to facilitate the surgery and LED placement[43,44]. The LED board spanned from 7 mm to 12 mm anterior to the interaural line, crossing from TEpd (dorsal posterior TE) to TEad (dorsal anterior TE) according to the Saleem and Logothetis atlas[45].

At each "stimulated" trial, one LED on the array was activated for 150 ms, and the LED power was kept constant during 150 ms of stimulation (square wave). The LED illumination levels used varied depending on the experiment and location on the cortex, but it was kept between 1 and 11 mW of total photometric output, adjusted to keep the animal's performance below the behavioral ceiling. The choice of LED and illumination power was kept constant at each perceptography cycle.

### Psychophysics

The experiments were performed in a well-lit test chamber in order to avoid retinal dark adaptation that could potentially help the animals detect the cortically delivered light through their skull (see Azadi et al. 2023 for more). The animals sat 57 cm away from a calibrated screen (32", 120 Hz, 1920 × 1080 IPS LCD, Cambridge Research System Ltd). The data was collected using a custom MWorks script[46] and a Mac Pro 2020. Eye tracking was performed using an Eyelink 1000 Plus (SR Research). All of the behavioral and surgical procedures used in this study were in accordance with the NIH guidelines.

### DaVinci

DaVinci, our illustrator engine, was built based on BigGAN[19], which generates images with high levels of naturalness, surpassing the other GANs. In order to construct the stimuli, DaVinci superimposed a random image over the seed image (both generated in BigGAN), then randomly perturbed the image parameters as well as the transparency of the top layer. The seed images were chosen randomly from 1000 classes provided by the BigGAN pretrained package. These images were pushed into the perceptography pipeline without any preselection. The altered image parameters included image class involvement (out of 1000 classes of ImageNet), truncation factor, and the z vector. Given our preliminary results (see Fig. 1d), we figured that most of the image search would happen not too far from the seed image; the two-layered image structure was considered to ease this. Nevertheless, we wanted DaVinci to be capable of venturing far and creating virtually any image by varying image parameters as well as layer transparency. To test this, we created seven target images that were not included in DaVinci's training set (ImageNet) and forced DaVinci to start from a random image seed and recreate the target image in an iterative process using a pixel dissimilarity loss function. The target images ranged from the picture of the dinner plate of one of the authors to modern art pieces warped in Photoshop. In all cases, DaVinci recreated the target

image with high fidelity (mean pixel similarity distance = 17.44%, StD = 4.28) (See Supplementary Fig S2).

## Ahab

Ahab was the optimizer that logged the behavioral responses and navigated DaVinci in order to find the perceptogram. The Ahab algorithm included a VGG-16[47,48] convolutional neural network (pretrained on ImageNet) as a feature extractor that kept track of the feature-vectors of the images that satisfy the following criteria: FA rate >50% and Miss rate <5%. By extracting and putting together the most common feature-vectors from the selected images, Ahab created an image prototype called average-feature-prototype (AFP). Then, Ahab created a pool of images sprayed around the AFP in the image space to achieve the range of image parameters in the vicinity of the AFP. Based on these parameters, Ahab guided DaVinci to make 2–6 mutants for each image.

## Feature distance

To measure feature-vector distance across images, we used a modified FID (Fréchet Inception Distance) measure (mseitzer/pytorch-fid package). This measure uses the feature-vector from multiple layers of its underlying deep neural network and is shown to strongly correlate with human visual quality judgments[49].

For comparing feature-vector distance across image classes, for each image class, we normalized the raw FID measure by the maximum FID observed in that class.

## Reporting summary

Further information on research design is available in the Nature Portfolio Reporting Summary linked to this article.

## Data availability

The data and material that support the findings of this study will be available upon the request. Pretrained model that we used in DaVinci and Ahab https://tfhub.dev/deepmind/biggan-deep-128/1. Seed_image collection: https://github.com/eliashahbazi/Seed_collection.git. Source data are provided with this paper.

## Code availability

The code is available upon request. Interested parties should contact Elia Shahbazi at elia.shahbazi@google.com for access.

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

## Acknowledgements

We thank Mark Eldridge and Reza Azadi for their crucial contributions to the surgeries. We thank Reza Azadi for the initial training of monkey Ph. This research was supported by the Intramural Research Program of the NIMH ZIAMH002958 (to A.A.)

## Author contributions

Concept: A.A. Experiment design: E.S. and A.A. Data collection and analysis: E.S. and T.M. Designing Ahab and DaVinci: E.S., M.P., and W.S. Writing the manuscript: A.A, E.S., and T.M.

## Funding

## Competing interests

The authors declare no competing interests.
