## [Peer Review File · Nature Communications]

Perceptography Unveils the Causal Contribution of Inferior Temporal Cortex to Visual Perception.REVIEWER COMMENTS

Reviewer #1 (Remarks to the Author):

In this manuscript, Shahbazi and colleagues investigated the nature of visual percepts elicited by optogenetic stimulation of the inferotemporal cortex. The authors trained macaque monkeys to detect brief (~150 ms) pulses of optogenetic stimulation in small (~1mm) regions of the central inferotemporal cortex while viewing GAN-generated images. Using a closed-loop machine learning procedure, the authors progressively perturbed these images to achieve a certain level of behavioral false-alarm rates from the animals. The outcome of this procedure was a collection of images (referred to as “perceptograms”) that the animals could not distinguish from optogenetic stimulation. The authors conclude that these “perceptograms” depict the visual percepts experienced by animals during optical stimulation of IT cortex.

The study is conceptually and technically novel: a “proof of principle” effort that opens new avenues for future studies investigating the links between neural activity and behavior. As such, it will likely make a valuable addition to the literature. However, the authors should address some concerns outlined below.

Major comments

(1) A major weakness of the manuscript is the lack of concurrent neurophysiological recordings—a point that the authors fully acknowledge in their Discussion. However, it seems that the authors could make some additional effort to link their behavioral findings to the known functional properties of the IT cortex. One way forward is to further investigate the nature of the image perturbations most commonly used by their machine learning procedure (perhaps focusing on the 15% of perceptograms scored as “natural”). Can the nature of these image perturbation parameters tell us something about IT encoding? Can we think about the Ahab optimized images as a description of neurons’ preferred stimulus? Why or why not?

(2) Another weakness of this manuscript is the writing style. The writing alternates between

being too informal and too jargonic. Additionally, the authors introduce new verbiage that may be unnecessary: for example, it seems the notion of a finding a “perceptogram” or performing “perceptography” is very similar to the notion of mapping a “projective field” (as cited in the Discussion). Even the descriptions of their image generation systems as “image illustrator” and “feature optimizer” seems unnecessary. The reference to “visual hallucinations” is also questionable. A careful re-write will likely improve readability and accessibility.

(3) How were “seed” images chosen? It would be useful for readers to know how seed images were determined to be sufficiently “natural looking” to be used in the experiments. Perhaps a supplemental figure could show chosen and unchosen seed images. Additionally, perhaps a supplemental figure include all seeds and all final “perceptograms”.

(4) The authors argue that the “nature and magnitude of perceptual effects depends on concurrent visual input”. Did the authors try delivering optical stimulation while the animal viewed a blank screen or a non-patterned stimulus (e.g. a black circle)? Do the authors know whether the animals can detect optical stimulation while looking at a blank screen or non-patterned stimulus?

(5) The authors argue that the change in false-alarm rates on day 1 of perturbing the seed image suggests that animals could not distinguish perturbed images from cortical stimulation. What makes the authors confident that the change in performance on day 1 isn’t simply due to the fact that the task is fundamentally different from the task performed on prior days in which only a static image was shown?

(6) Why was optogenetic stimulation always delivered concurrently with a perturbed image instead of with the seed image? Wouldn’t that be a reasonable thing to do? It would be good to clarify this point to readers.

(7) How long does the perceptogram last? Have the authors tried repeating the same procedure for the same LED site within a single session or across consecutive days? Additionally, did both animals evolve percepts of equal/different complexity?

(8) Could the Opto Array be used to estimate the region of vector expression?

(9) Figure 1B: Why do both animals show a strong dip in false-alarm rate at trials near 1000?

(10) Figure 1D: Don't we expect a monotonic relationship between the magnitude of image perturbation and the false alarm rate? For depicting distance from seed class, can the authors provide examples for a single seed image instead of different ones?

(11) Figure 5A: The description of this panel in the main text and the figure legend is unclear. Can the authors show the actual images alongside the "heat maps"?

Minor comments

(1) Viral vector methods should be expanded. What was the titer of the viral vector? What was the total volume of vector injected? How long after the injections did experiments start? What's the wavelength of light illumination?

(2) It is unclear why the authors chose to target CIT (instead of AIT, PIT or V4)?

(3) It is unclear what is "high-throughput" about the authors' approach?

(4) The manuscript title seems quite broad/generic, especially given the number of other studies that have investigated "the causal role of IT cortex in visual perception".

(5) The references linked to the following sentence are incorrect: Technical details about the Opto-Array and relevant surgical protocols can be found in our earlier reports (4,13).

References 4 and 13 do not refer to the Opto-Array.

Reviewer #2 (Remarks to the Author):

This study uses a powerful new approach to reveal the perceptual consequences of direct stimulation of the inferotemporal cortex (IT). Monkeys were trained to report a brief (150 ms) pulse of light to C1V1-expressing IT neurons. An image, nominally irrelevant to the monkeys' behavior, was presented for 1 s and modified during the stimulation pulse. On a randomly selected subset of the trials, this image manipulation was optimized to maximize the false alarm rate (while maintaining low miss rates on no-stimulation trials). This optimization was achieved using cutting-edge machine learning techniques. The image manipulation that caused the monkey to report having been optically stimulated, termed the "perceptogram", is a reasonable estimate for the perceptual consequences of the combined effects of image manipulation and optical stimulation of IT.

This is exciting work. The perceptogram approach is a powerful one that has utility beyond the particular application used in this study. The fact that the Opto-Array does not allow simultaneous recording is regrettable but does not detract from my enthusiasm about it or this study. The fact that it allows repeated optogenetic manipulations across many days is a major strength.

Major comments

Several metrics were used to quantify differences between images (Frechet Inception Distance, pixel distance, RMSE, normalized feature distance). Why one metric was chosen over another for any particular analysis was unclear.

My understanding is that the Frechet Inception Distance (FID) is appropriate for comparing (presumed Gaussian) distributions of images (represented in the output space of the network). In some analyses, however, the FID was used to measure distances between pairs of individual images (e.g. Fig. 2b and 2c). Is the FID in this case just the sum of squared differences between the output-layer representations of the images? Why do FID values span two orders of magnitude across analyses (e.g. Fig. 3b and 3c)?

Minor comments

The abstract asserts that the animals "could not discriminate" the perceptograms from the state of being cortically stimulated. It would be more accurate to say that they "did not discriminate" these events.

Figure 1b: In the session depicted, how did the images change across trials? Was Ahab engaged? The correlation between traces from the two monkeys is surprising. What accounts for the steep drop in false alarm rate at trial ~1000 (for both monkeys) and at ~1600 (for monkey Sp) and ~1800 (for monkey Ph)?

Figure 1d: How many (non-stimulated) trials are in each distance bin? This is important for interpreting the proportion of false alarms in each bin.

Figure 1d plots proportion of false alarms as a function of "Distance from seed class" (the legend refers to this quantity as "normalized feature distance"). How is this quantity calculated? This is important for interpreting the non-monotonicity of false alarm rate. Might the false alarm rate be non-monotonic in "normalized feature distance" but monotonic in some other equally reasonable image distance metric? If so, how are we to interpret the non-monotonicity?

"...the images selected independently by the animals' behavior across families share increasingly more features as the process continues." I think this statement refers to the convergence of the FID, but the way it is phrased suggests that the number of features was tabulated and compared.

Figure 3a: I assume that this analysis assumes a stationary false alarm rate. Is this correct? If so, how reasonable is this assumption? Do the results change if a non-stationary false alarm rate is assumed?

Figure 3c. The fact that the two perceptograms converge in FID does not necessarily indicate that they appear similar to a human observer, does it? Please show the two perceptograms so the reader can compare them.

"...we showed image perturbations of one seed image (150ms) temporally sandwiched between images of another seed". Did this manipulation affect the hit rate?

A citation to Schalk et al. 2017 in the sentence that begins with "As reported in our earlier work..." seems irrelevant. Perhaps a citation to Azaidi et al. 2023 was intended here?

"their net effect is not expected to drift far from the seed image". This sentence is difficult to parse because an effect and an image do not occupy the same space (so they can neither drift towards nor away from each other). What does "net effect" mean here, exactly?

Figure 4a: Are the data pooled across monkeys? If so, is the effect consistent for both?

The panels on the left of Figure 5a (low stimulation intensity) have low contrast, which makes them difficult to compare with the panels on the right. Boosting the contrast (and showing the seeds and the perceptograms) would help the reader to understand the result.

Figure 5b: Is RMSE calculated on pixel values?

Figure 5d: Is behavior (hits, misses, false alarms, correct rejections) on low power trials significantly different from behavior on no-stimulation trials? Which two of the six stimulation intensities were included in the data shown? On what basis were these selected?

"...for "a" part of shape space." Why is "a" in quotation marks?

Does the false alarm rate produced by the perceptograms vary with the anterior/posterior position of the stimulating LED?

Figure 6b: How is "pixel distance" defined?

Methods: Azadi et al. reference if #6 not #4.

Authors are not listed for reference #22.

Reviewer #3 (Remarks to the Author):

Shahbazi and his colleagues describe experiments exploring the percepts produced by optogenetically exciting neurons in the inferotemporal visual cortex (IT) of macaque monkeys. Previous studies from this group and others have shown that monkeys can detect artificial microstimulation of IT, and characterized how concurrent visual stimulus presentations can affect the animal's ability to detect the microstimulation. What is new here is the use of a generative adversarial neural network (GAN) to sequentially and systematically modify visual images to minimize the animal's ability to distinguish the visual image from the optogenetic perturbation. GANs have been used previously to modify visual images to maximize the responses of individual visual neurons, but have not been used to match a visual stimulus to an induced percept. The manuscript is interesting both for the new results it presents and for the new approach it lays out. There are nevertheless several important aspects of the experiments and results that need to be made clear before the results can be readily evaluated.

Major Comments

1) Because the perturbed images were presented briefly and with no gaps separating them from the unperturbed images, the sequence would have produced robust apparent motion. It would be valuable to analyze the motion energies produced by the different effective and ineffective stimuli for a given cortical stimulation site to see whether motion-based masking of the optogenetic stimulation can be ruled out in favor of some form perception.

2) Figure 5: One of the important observations in this manuscript is that the intensity of cortical optogenetic stimulation systematically alters the nature of the image that has the greatest effect on behavioral performance. Inexplicably, however, when they present this result the authors switch to new measures (RMSE, Miss rate, heat maps -- without a scale -- and example images). These are completely incommensurate with those used up to this point (FA rate, FID to seed, or "FID to perceptogram"). These results should be presented

using measurements that can be directly compared with the previous data.

3) Behavioral Measures:

The authors made good use of signal detection theory, but omitted a few important measures.

3a) Page 5, Figure 2: The interpretation of the shift in FA rate with exposure to the optimized perturbed images depends critically on those images remaining a constant fraction of all perturbed images. Was that the case? What was the overall fraction of the stimulus presentations involving the optimized stimulus versus all others?

3b) Page 3: How was the miss rate kept fixed while increasing FA and not providing differential rewards for hits and correct rejections?

4) Unjustified claims:

4a) Page 3/Figure 1: The authors argue that the rapid rise in FAs when visual perturbations are introduced argues that the animals perceive the optogenetic stimulation as visual. The 50% FA rates suggest the animals initially generalized their cortical perturbation detection task to be a general perturbation detection task, such that all visual perturbations would be rewarded perturbation. After a few hundred trials, they had an “ah-ha” moment and stopped responding to the visual perturbations (which they presumably could distinguish throughout). Generalizing to other perturbations does not provide strong support for IT stimulation producing a visual experience.

4b) The authors claim that these images reveal and depict the contents of the complex hallucinatory percepts induced by neural perturbation in IT. That is an overstatement. They find that radically different images can substitute for optogenetic stimulation of a given cortical site (cf. the left 2 image columns or the right 4 image columns in Figure 5c). We can infer that each of these image groups captures some important component of the percept produced by IT stimulation, but none can be said to reveal or depict the contexts of that

percept, or a “picture” of visual hallucination.

4c) Page 1: The statement that animals “could not discriminate [the images] from the state of being cortical stimulated” needs to be qualified. Performance with optogenetic stimulation was clearly distinguishable in the animals’ behavior.

4d) Page 1: The design of these experiments does not match standard definitions of “closed-loop” stimulation.

4e) Page 1: The claims that these observations “enable” better visual prosthetic devices or relate directly to hallucinations in mental disorders also seem difficult to defend.

5) Missing details:

5a) Was the optogenetic stimulation appropriately delayed for IT neuronal latencies?

5b) Page 15: “Lateral bank of the central IT cortex” does not narrow things down well enough for other to replicate these experiments.

5c) Page 15: Presumably the LED power was constant (not pulsed) during the 150 ms stimulation period, but this should be stated.

5d) Figure 6: The x-axes should be given in millimeters (or ranges of millimeters).

6) Page 15: A statement in the Methods suggests that the current results are based at least in part on data that have been previously published. There is no problem with reusing data, but the manuscript must include a statement making clear where the data have previously appeared, whether the current data overlap completely or in part, and whether any of the current results have been reported previously.

7) Given that these two subjects were capable of detecting optogenetic stimulation in the absence of any visual stimulus, it would be helpful for the authors to discuss whether less

biased images might be produced starting from a blank slate.

Minor Comments

8) I think it is up to the authors to decide whether they want to attach themselves to a newly forged label, however “perceptograms” seems a bit desperate given how these images likely to relate to percepts (Comment 1).

9) Page 15: “10 uLit” is not SI. The number of viral particles injected should be provided.

10) Pages 10-12: The material in “Perceptograms and natural image manifolds” would be more naturally presented in the Discussion.

11) Page 5: It is inappropriate to speak of “untraining” the monkeys in a signal detection theory task. What the authors probably mean is that the animals will adjust their behavioral criterion in response to sustained challenges from particular perturbed image sets.

12) The y-axis in Figure 3a should start at zero. It is graphically misleading to do otherwise.

13) The title is not informative about the experiments performed and seems to offer more than the data can provide.

14) Referring to the visual stimulus as a “video” encourages readers to incorrectly believe that the 1 s video was dynamic throughout. It would be safer to refer to it as 1 s sequence of static images.

Reviewer #1 (Remarks to the Author):

In this manuscript, Shahbazi and colleagues investigated the nature of visual percepts elicited by optogenetic stimulation of the inferotemporal cortex. The authors trained macaque monkeys to detect brief (~150 ms) pulses of optogenetic stimulation in small (~1mm) regions of the central inferotemporal cortex while viewing GAN-generated images. Using a closed-loop machine learning procedure, the authors progressively perturbed these images to achieve a certain level of behavioral false-alarm rates from the animals. The outcome of this procedure was a collection of images (referred to as “perceptograms”) that the animals could not distinguish from optogenetic stimulation. The authors conclude that these “perceptograms” depict the visual percepts experienced by animals during optical stimulation of IT cortex.

The study is conceptually and technically novel: a “proof of principle” effort that opens new avenues for future studies investigating the links between neural activity and behavior. As such, it will likely make a valuable addition to the literature. However, the authors should address some concerns outlined below.

We thank the reviewer for the encouraging assessment of our work.

Major comments

(1) A major weakness of the manuscript is the lack of concurrent neurophysiological recordings—a point that the authors fully acknowledge in their Discussion. However, it seems that the authors could make some additional effort to link their behavioral findings to the known functional properties of the IT cortex. One way forward is to further investigate the nature of the image perturbations most commonly used by their machine learning procedure (perhaps focusing on the 15% of perceptograms scored as “natural”). Can the nature of these image perturbation parameters tell us something about IT encoding? Can we think about the Ahab optimized images as a description of neurons’ preferred stimulus? Why or why not?

We thank the reviewer for pushing us to talk more about possible neural underpinnings of the current findings. Originally, we refrained from doing so because of its speculative nature in the absence of actual recording data. But we agree with the reviewer that more shall be discussed here.

What is the relationship between perceptograms and the preferred stimuli of their driving neurons? It is a tantalizing question. “Preferred stimuli” of neurons reveal how the visual signal is encoded in IT cortex, and Perceptograms show how the signal gets decoded from IT by the rest of the brain. These two do not necessarily match, and the relationship between the two can vary under different decoding frameworks. In some cases of sensory processing, e.g., in early somatosensory processing, neurons are tightly tuned to specific physical stimuli. Activation of such a neuron induces the appearance of its related sensory stimulus in perception. Such a

direct one-to-one hypothetical relationship between the preferred stimulus of a sensory neuron and the percept it arises is known as the labeled line hypothesis. Alternatively, more complex decoding frameworks also exist. For instance, a medium wavelength cone on the retina responds mostly to the green light, but its activation does not necessarily induce perception of the color green. The perceived color, in this case, depends on the activation ratio of other cone types as well as the position of the activated cone in the retinal cone mosaic (Hofer et al., 2005). Now, is decoding in IT cortex like a labeled line or like the retinal cones? Our results are not consistent with the former. If IT neurons were labeled lines, we expected to see similar features in the perceptograms that come from a single channel independent of the base image. However, we found that the base image, which in turn determines the activity pattern of other IT neurons (like the case of retinal color), massively influences the outcome of local neural perturbation in IT. An analysis of the perceptograms using Yolo (a real-time object detection system) revealed that 82% of the features in the perceptograms are shared (StD=21%) with their corresponding seed images. The analysis also showed that the added features (compared to the seed) of the perceptograms acquired from a single channel have only a little in common with each other (0% and 7% in Sp and Ph, respectively), which is not different (t-test, $p>0.4$ for both animals) from the overlap between the added features obtained from different channels (0% and 10% in Sp and Ph respectively).

This suggests that the pattern of neural activity in the cortex (which varies by the base image) has a strong influence on the outcome of local stimulation in IT cortex. This, in turn, means that, most probably, the labeled line framework is not sufficient to explain the mapping between neural activity in IT cortex and object perception. If not the labeled line hypothesis, what framework can explain decoding in IT? A systematic search for the solution requires measuring both; the projective fields of neurons (via Perceptograms) as well as their response properties of neurons (via neural recording). Here we have provided the proof of concept for the first leg of the solution and hope to soon overcome the technical challenges on the way of neural recordings. Needless to say, given the gravity of this task, we find it bigger than the capacity of one lab and hope these results inspire our colleagues in other areas of sensory processing to join the effort.

These discussions and the new analysis are added to the manuscript now (page 9). We hope the reviewer finds them as an improvement.

(2) Another weakness of this manuscript is the writing style. The writing alternates between being too informal and too jargonic. Additionally, the authors introduce new verbiage that may be unnecessary: for example, it seems the notion of a finding a “perceptogram” or performing “perceptography” is very similar to the notion of mapping a “projective field” (as cited in the Discussion). Even the descriptions of their image generation systems as “image illustrator” and “feature optimizer” seems unnecessary. The reference to “visual hallucinations” is also questionable. A careful re-write will likely improve readability and accessibility.

We have now carefully revised the writing style. We have dropped the notion of the “Projective field” to avoid redundancy. The terms “image illustrator” and “feature optimizer” reflect technical

terminology used in computer vision (Lin et al. 2015, Riyahi et al. 2022, Ku et al. 2009), so we kept them. The term “visual hallucination” is now clearly defined in the manuscript (page 1).

1- Lin, Y.H. and Tsai, M.S., 2015. The integration of a genetic programming-based feature optimizer with fisher criterion and pattern recognition techniques to non-intrusive load monitoring for load identification. *International journal of green energy*, 12(3), pp.279-290.

2- Riyahi, M., Rafsanjani, M.K., Gupta, B.B. and Alhalabi, W., 2022. Multiobjective whale optimization algorithm-based feature selection for intelligent systems. *International Journal of Intelligent Systems*, 37(11), pp.9037-9054.

3- Ku, C.W., Tsai, K.L. and Li, P.Y., 2009, October. Medical Images Illustrator: A Flexible Image Processing System. In *Proceedings: APSIPA ASC 2009: Asia-Pacific Signal and Information Processing Association, 2009 Annual Summit and Conference* (pp. 887-894). Asia-Pacific Signal and Information Processing Association, 2009 Annual Summit and Conference, International Organizing Committee.

(3) How were “seed” images chosen? It would be useful for readers to know how seed images were determined to be sufficiently “natural looking” to be used in the experiments. Perhaps a supplemental figure could show chosen and unchosen seed images. Additionally, perhaps a supplemental figure include all seeds and all final “perceptograms”.

We thank the reviewer for demanding more clarity here. The seed images were chosen randomly from 1000 classes provided by the BigGAN pretrained package. These images were pushed into the perceptography pipeline without any pre-selection. The images were all subject to random image mutations (by DaVinci), and the ones that created FAs survived for the next round.

Regarding the naturalness of these images, BiGGAN, the engine at the core of DaVinci, is shown to generate images with high levels of naturalness, surpassing the other GAN models (Brock et al. 2018).

These points are now clarified in the manuscript (page 12).

Supplementary Figure 2 includes the full collection of the perceptograms and their seeds.

Brock, Andrew, Jeff Donahue, and Karen Simonyan. "Large scale GAN training for high fidelity natural image synthesis." *arXiv preprint arXiv:1809.11096* (2018)

(4) The authors argue that the “nature and magnitude of perceptual effects depends on concurrent visual input”. Did the authors try delivering optical stimulation while the animal viewed a blank screen or a non-patterned stimulus (e.g. a black circle)? Do the authors know whether the animals can detect optical stimulation while looking at a blank screen or non-patterned stimulus?

This is indeed a very important question. In fact, we have already dedicated two very recent papers to the study of this question (Azadi et al., 2023, Lafer-Sousa et al., 2023). In sum, we found that detecting cortical stimulation is difficult for the animals when looking at a blank screen. While the animals still perform the task above the chance level, introducing a visual stimulus dramatically increases their performance. Blending the visual stimulus into the background by reducing its contrast/visibility or its size takes a big toll from detection performance. This is consistent with a human study (Murphy et al. 2009) showing that humans barely notice stimulation of their fusiform cortex when blindfolded.

Given the lower dynamic range of performance for the blank screen as well as the technical complications of mutating blank images, we avoided perceptography with a blank seed for this initial study. We aim to explore this complicated matter systematically in our following studies.

These points are now reflected in the manuscript (page 3).

- 1- Azadi, R., Bohn, S., Lopez, E., Lafer-Sousa, R., Wang, K., Eldridge, M.A. and Afraz, A., 2023. Image-dependence of the detectability of optogenetic stimulation in macaque inferotemporal cortex. *Current Biology*.
- 2- Lafer-Sousa, R., Wang, K., Azadi, R., Lopez, E., Bohn, S. and Afraz, A., 2023. Behavioral detectability of optogenetic stimulation of inferior temporal cortex varies with the size of concurrently viewed objects. *Current Research in Neurobiology*, 4, p.100063.
- 3- Murphey, D.K., Maunsell, J.H., Beauchamp, M.S. and Yeshor, D., 2009. Perceiving electrical stimulation of identified human visual areas. *Proceedings of the National Academy of Sciences*, 106(13), pp.5389-5393.

(5) The authors argue that the change in false-alarm rates on day 1 of perturbing the seed image suggests that animals could not distinguish perturbed images from cortical stimulation. What makes the authors confident that the change in performance on day 1 isn't simply due to the fact that the task is fundamentally different from the task performed on prior days in which only a static image was shown?

This is a great question; thanks for asking. If the FA rate is increased as a result of unfamiliarity with the task (which in signal detection theory language translates to more "task difficulty"), we expect the animal's Miss rate also to increase. However, as the updated figure (below) shows, the Miss rate remains constant and low.

We have updated the figure in the paper (Fig 1) and clarified the claims (page 3).

(6) Why was optogenetic stimulation always delivered concurrently with a perturbed image instead of with the seed image? Wouldn't that be a reasonable thing to do? It would be good to clarify this point to readers.

We thank the reviewer for demanding more clarity here. Please note that if we show the perturbed images only for the non-stimulated trials, we will provide a huge cue for the monkey to perform the task without paying attention to cortical stimulation. To avoid this problem, we presented the perturbed images in both stimulated and non-stimulated trials. This way, the animals cannot use any clue other than brain stimulation to discriminate between the two types of trials. Also note that perturbed images included many examples that were very close to the seed images, making the "only seed" condition a subset of the initial pool.

This point is now clarified in the manuscript (page 6).

(7) How long does the perceptogram last? Have the authors tried repeating the same procedure for the same LED site within a single session or across consecutive days? Additionally, did both animals evolve percepts of equal/different complexity?

This is a great question. The perceptography process takes about a month of data collection for each seed (~40K trials). As a result, it is not possible to redo it in a single session. But, we have performed the entire process for a given seed and LED/intensity twice for each monkey. There were 24 and 10 days between the procedures for the monkeys, respectively. Taking into account the length of the second round of perceptography, there was more than a month between the first and the second perceptogram in each animal. The perceptograms looked identical (see Fig 3. C). Therefore, we can assure that the perceptograms remain unchanged, at least for the period of ~1 month.

As for the complexity of perceptograms in the two monkeys, following we have provided a quantitative comparison; the Y axis is the normalized FID distance from the seed image. This is a feature distance measure; higher values indicate more additional features, thus more complexity. We found no difference in the complexity of perceptograms in the two animals.

(8) Could the Opto Array be used to estimate the region of vector expression?

Yes. It is possible to use the Opto Array to estimate the region of vector expression. Basically, the performance is expected to be low for the areas with low expression. We have already shown that detection of the LED illumination is not possible without vector expression (Azadi et al., 2023). Although, this is an indirect and expensive way of testing the expression. In typical practice, we darken the OR before array implantation and use UV light to determine the extent of vector expression. The appropriate citations for the surgical method are now reported in the paper (pages 11-12).

(9) Figure 1B: Why do both animals show a strong dip in false-alarm rate at trials near 1000?

Sharp eyes! The data for the perceptography training task was collected over two days. The dip around trial 1000 indicates the disruption induced by the transition from day 1 to day 2 of data collection. An indicator representing data collection days is added to the image now (Figure 1. B).

(10) Figure 1D: Don't we expect a monotonic relationship between the magnitude of image perturbation and the false alarm rate? For depicting distance from seed class, can the authors provide examples for a single seed image instead of different ones?

No! In fact, it is exactly what should not happen. If the FA rate increases monotonically by increasing the image perturbation magnitude, it will indicate that the more we change an image, the more likely it is for the monkey to FA. In other words, it would indicate that the monkey is sensitive simply and only to the non-specific amount of image perturbation. The non-monotonic

relationship observed here reassures us that the animal is not fooled by the mere magnitude of image perturbation. Instead, the animal is looking for a specific image, and the perturbation magnitude should match that, not more, not less. Given the wide range of image manipulations used in this study, we expect our target image perturbation to land in the middle of the perturbation range.

This point is now explained in the manuscript (page 4). Also, we have modified the figure and provided examples from a single seed image (Figure 1.D).

(11) Figure 5A: The description of this panel in the main text and the figure legend is unclear. Can the authors show the actual images alongside the “heat maps”?

Thanks. The figure is now updated according to the reviewer’s suggestion (Figure 5. a).

Figure 5b: Is RMSE calculated on pixel values?

Yes. RMSE calculates the average of the squared differences between corresponding pixels in the two images and then takes the square root of this value.

Minor comments

(1) Viral vector methods should be expanded. What was the titer of the viral vector? What was the total volume of vector injected? How long after the injections did experiments start? What’s the wavelength of light illumination?

We thank the reviewer for demanding more clarity here. The details are mentioned in the papers cited in the manuscript (Azadi et al., 2023).

We injected AAV5-CaMKIIa-C1V1(t/t)-EYFP (nominal titer: 8×10^{12} particles/ml) into the cortex using an injection array of four 31-gauge needles arranged in a 2×2 mm square. We tiled the central IT cortex four times with sixteen evenly spaced injection sites, resulting in a 6 mm x 6 mm viral expression area. At each injection site, ten μ l of the virus were injected at a rate of 0.5 μ l/min, totaling an injection volume of 160 μ L. After each injection, the array was removed, and a ten-minute wait period was allowed for virus diffusion into the cortical tissue to ensure uniform viral expression and reduce anesthesia-controlled time. After finishing all injections, we closed the animals and waited an entire month before a second surgery for array implantation. The training typically started a month after the second surgery, followed by data collection sessions. The LEDs used in this experiment were green (530 nm).

To make the important details available to the reader, we have now extended and clarified the discussion about viral injection and viral vector methods (pages 11-12).

(2) It is unclear why the authors chose to target CIT (instead of AIT, PIT, or V4)?

The goal was to get as close as possible to anterior IT (the highest level of IT processing). CIT is as close as we could get, given the surgical limitations. We are now performing separate projects on V4 and V1.

(3) It is unclear what is “high-throughput” about the authors’ approach?

The high-throughput points to the massive number of image presentations (~20K for each perceptogram on average). The data obtained here is the result of processing the behavioral responses of the animals to this high number of trials for each seed image / cortical site.

(4) The manuscript title seems quite broad/generic, especially given the number of other studies that have investigated “the causal role of IT cortex in visual perception”.

We thank the reviewer for the suggestion. The title is now modified. (title)

(5) The references linked to the following sentence are incorrect: Technical details about the Opto-Array and relevant surgical protocols can be found in our earlier reports (4,13). References 4 and 13 do not refer to the Opto-Array.

We apologize for the error. The citation is fixed.

Reviewer #2 (Remarks to the Author):

This study uses a powerful new approach to reveal the perceptual consequences of direct stimulation of the inferotemporal cortex (IT). Monkeys were trained to report a brief (150 ms) pulse of light to C1V1-expressing IT neurons. An image, nominally irrelevant to the monkeys' behavior, was presented for 1 s and modified during the stimulation pulse. On a randomly selected subset of the trials, this image manipulation was optimized to maximize the false alarm rate (while maintaining low miss rates on no-stimulation trials). This optimization was achieved using cutting-edge machine-learning techniques. The image manipulation that caused the monkey to report having been optically stimulated, termed the "perceptogram", is a reasonable estimate for the perceptual consequences of the combined effects of image manipulation and optical stimulation of IT.

This is exciting work. The perceptogram approach is a powerful one that has utility beyond the particular application used in this study. The fact that the Opto-Array does not allow simultaneous recording is regrettable but does not detract from my enthusiasm about it or this study. The fact that it allows repeated optogenetic manipulations across many days is a major strength.

We thank the reviewer for the encouragement and the constructive comments.

Major comments

Several metrics were used to quantify differences between images (Frechet Inception Distance, pixel distance, RMSE, normalized feature distance). Why one metric was chosen over another for any particular analysis was unclear.

My understanding is that the Frechet Inception Distance (FID) is appropriate for comparing (presumed Gaussian) distributions of images (represented in the output space of the network). In some analyses, however, the FID was used to measure distances between pairs of individual images (e.g. Fig. 2b and 2c). Is the FID in this case just the sum of squared differences between the output-layer representations of the images? Why do FID values span two orders of magnitude across analyses (e.g. Fig. 3b and 3c)?

The FID score is a measure that compares the statistical properties of feature vectors in two sets using the Inception V3 model (here, we used the last layer). Each set may contain features from multiple images or a single image (Deaconu, G. 2021). Compared to other standard distance measures (e.g., RMSE), FID provides a distance metric sensitive to features rather than pixels; this is why it makes an appropriate measure of high-level similarity/dissimilarity between groups of images or individual image pairs.

FID ranges from 0 to infinity. As for Figure 3b, since we averaged different groups of seed-altered images, we used the normalized FID distance. However, we forgot to name the axis label properly. We thank the reviewer for noticing this error. The figure label is now fixed.

Deaconu, G. (2021, November 16). *Assessing image similarity using inception V3 and FID score*. Towards Data Science. <https://towardsdatascience.com/assessing-similarity-between-two-images-groups-using-inception-v3-and-fid-score-4b0367a74e67>

Minor comments

The abstract asserts that the animals "could not discriminate" the perceptograms from the state of being cortically stimulated. It would be more accurate to say that they "did not discriminate" these events.

The reviewer is right; the text is updated with the reviewer's suggestion.

Figure 1b: In the session depicted, how did the images change across trials? Was Ahab engaged? The correlation between traces from the two monkeys is surprising. What accounts for the steep drop in false alarm rate at trial ~1000 (for both monkeys) and at ~1600 (for monkey Sp) and ~1800 (for monkey Ph)?

In Figure 1. b, the first 500 trials come from the last data collection session with solid images (before introducing the dynamic image perturbation). The rest of the data show the results from the first two days of training following the introduction of the dynamic image change. The drop at ~1000 aligns with the transition between the two days. Both monkeys had better performance at the end of the first day; this is the dip observed around trial 1000 in both animals. Both animals eventually learned the task on the second day, and that caused the final dramatic drop in FA rate.

Figure 1d: How many (non-stimulated) trials are in each distance bin? This is important for interpreting the proportion of false alarms in each bin.

We assigned all conditions (including the magnitude of image perturbation) randomly and in a balanced design. Each bin contains 440-470 non-stimulated trials in Figure 1d. This is now mentioned in the figure legend.

Figure 1d plots proportion of false alarms as a function of "Distance from seed class" (the legend refers to this quantity as "normalized feature distance"). How is this quantity calculated? This is important for interpreting the non-monotonicity of false alarm rate. Might the false alarm rate be non-monotonic in "normalized feature distance" but monotonic in some other equally reasonable image distance metric? If so, how are we to interpret the non-monotonicity?

The normalized feature distance comes from the BigGAN interpolation factor. This indicates the proportion of out-of-class features mixed in for each seed provided by the generative engine. Here, 0 indicates that all the features are from the seed class, and one means that all the features come from other classes (Figure 1.d legend).

Changing the distance measure does not affect the overall shape of the distribution. Following, we provide the false alarm rate distribution over the pixel distance.

"...the images selected independently by the animals' behavior across families share increasingly more features as the process continues." I think this statement refers to the convergence of the FID, but the way it is phrased suggests that the number of features was tabulated and compared.

We appreciate the reviewer for bringing this point to our attention. The statement refers to the convergence of FID and the similarity between the images selected independently by the animals' behavior across families as the process continues. The statement is now rephrased in the paper for more clarity (page 5).

Figure 3a: I assume that this analysis assumes a stationary false alarm rate. Is this correct? If so, how reasonable is this assumption? Do the results change if a non-stationary false alarm rate is assumed?

Yes, we assumed a stationary false alarm rate here. This assumption comes from the observation that the false alarm rate remains relatively constant for DaVinci images, reflecting the animals' high performance on the task. Figure 2a shows this. A simple one-way ANOVA supports this assumption as there is no effect of the session on the FA rate. P-values of 0.82 and 0.92 (df= 6 and 7 for Sp and Ph, respectively).

Figure 3c. The fact that the two perceptograms converge in FID does not necessarily indicate that they appear similar to a human observer, does it? Please show the two perceptograms so the reader can compare them.

Thanks for the wonderful suggestion. The first and second perceptograms, as well as some transitional images, are now shown. We have shown this only for one monkey to avoid cluttering the image (Fig 3. c).

"...we showed image perturbations of one seed image (150ms) temporally sandwiched between images of another seed". Did this manipulation affect the hit rate?

The hit rate didn't change compared to the regular hit rate of perceptograms and was 98% and 100% for Sp and Ph, respectively.

A citation to Schalk et al. 2017 in the sentence that begins with "As reported in our earlier work..." seems irrelevant. Perhaps a citation to Azaidi et al. 2023 was intended here?

We apologize for the error; the citation has been fixed in the paper.

"their net effect is not expected to drift far from the seed image". This sentence is difficult to parse because an effect and an image do not occupy the same space (so they can neither drift towards nor away from each other). What does "net effect" mean here, exactly?

Thanks for demanding more clarity here. We did not mean averaging the brain stimulation and the image; we meant that the average of all perturbed images (excluding the effect of brain stimulation) is centered around the seed image. In other words, each image perturbation (not to be mixed with brain perturbation) drifts the seed image into an arbitrary direction. Still, given that they are random and mostly subtle, these drifts remain centered at the seed image.

We have now clarified the text to better communicate this point (page 6).

Figure 4a: Are the data pooled across monkeys? If so, is the effect consistent for both?

Yes, the data are pooled across the two monkeys for Figure 4.a. A separate analysis of the data shows identical trends in both monkeys (following).

The panels on the left of Figure 5a (low stimulation intensity) have low contrast, which makes them difficult to compare with the panels on the right. Boosting the contrast (and showing the seeds and the perceptograms) would help the reader to understand the result.

We thank the reviewer for pointing out this problem. The figure is now revised to make it clear and easy to understand.

Figure 5b: Is RMSE calculated on pixel values?

Yes. It is now clarified in the manuscript (figure 5).

RMSE calculates the average of the squared differences between corresponding pixels in two images and then takes the square root of this value.

Figure 5d: Is behavior (hits, misses, false alarms, correct rejections) on low power trials significantly different from behavior on no-stimulation trials? Which two of the six stimulation intensities were included in the data shown? On what basis were these selected?

We thank the reviewer for demanding more clarity on these points.

As for the behavior, there was a very large difference between low-power and no-stimulation trials: most of the low-power trials were categorized as stimulated trials, and most of the non-stimulated trials were reported as so by the animals. As the figure below shows, behavioral performance in both low-power and high-power trials remained far above chance, indicating that low-power trials were well discriminated from no stimulation trials by the animals. This plot shows data averaged for both monkeys, but each monkey shows the same trend. Also, as the bar plot in Figure 5.d shows, even though the miss rate in low-power trials is higher than in the high-power trials, it is relatively low in both monkeys, indicating high discriminability of the low-power trials from no stimulation.

As for the six stimulation intensities, we apologize for the miscommunication. There were only two stimulation intensities (in each monkey) in the experiment concerning high and low-power perceptography. The intensity levels for each animal were chosen to keep the performance high but below the behavioral ceiling. This sums up to 4 stimulation intensities for this experiment: Monkey Sp: Low power=1 mW High power=2 mW, Monkey Ph: Low power=1 mW High power=3 mW. As described in the paper, occasionally, we had to adjust the stimulation intensity before each round of perceptography in a new channel/monkey to keep the performance high but below the ceiling. This summed the total of stimulation intensities used in our experiments to 6. We have clarified the manuscript to avoid this confusion (page 7).

"...for "a" part of shape space." Why is "a" in quotation marks?

Thanks. Fixed. (page 7)

Does the false alarm rate produced by the perceptograms vary with the anterior/posterior position of the stimulating LED?

Great question. The FA rate in different locations remains statistically unchanged. ($p > 0.05$)

Figure 6b: How is "pixel distance" defined?

Pixel distance is RMSE, indicating the square root of the average of the squared differences between corresponding pixels in two images.

Methods: Azadi et al. reference if #6 not #4.

Thanks. Fixed.

Authors are not listed for reference #22.

Thanks. Fixed.

Reviewer #3 (Remarks to the Author):

Shahbazi and his colleagues describe experiments exploring the percepts produced by optogenetically exciting neurons in the inferotemporal visual cortex (IT) of macaque monkeys. Previous studies from this group and others have shown that monkeys can detect artificial microstimulation of IT, and characterized how concurrent visual stimulus presentations can affect the animal's ability to detect the microstimulation. What is new here is the use of a generative adversarial neural network (GAN) to sequentially and systematically modify visual images to minimize the animal's ability to distinguish the visual image from the optogenetic perturbation. GANs have been used previously to modify visual images to maximize the responses of individual visual neurons, but have not been used to match a visual stimulus to an induced percept. The manuscript is interesting both for the new results it presents and for the new approach it lays out. There are nevertheless several important aspects of the experiments and results that need to be made clear before the results can be readily evaluated.

We thank the reviewer for the constructive and thorough review of our work.

Major Comments

1) Because the perturbed images were presented briefly and with no gaps separating them from the unperturbed images, the sequence would have produced robust apparent motion. It would be valuable to analyze the motion energies produced by the different effective and ineffective stimuli for a given cortical stimulation site to see whether motion-based masking of the optogenetic stimulation can be ruled out in favor of some form perception.

This is an excellent comment; we thank the reviewer for demanding more clarity here. We calculated the motion energy for Perceptoram sequences as well as 250 randomly chosen sequences from the initial DaVinci pool. To calculate motion energy, we used Python's "pymoten" protocol which is a standard package for motion energy measurement (Nunez-Elizalde AO, Deniz F, Dupré la Tour T, Visconti di Oleggio Castello M, and Gallant JL (2021). pymoten: a scientific python package for computing motion energy features from video. Zenodo.).

As the following plot demonstrates, we observed no difference in the motion energy content of the two groups ($t(279) = 0.11$, $p = 0.46$). This result and short discussion are now reflected in the manuscript (page 7).

2) Figure 5: One of the important observations in this manuscript is that the intensity of cortical optogenetic stimulation systematically alters the nature of the image that has the greatest effect on behavioral performance. Inexplicably, however, when they present this result the authors switch to new measures (RMSE, Miss rate, heat maps -- without a scale -- and example images). These are completely incommensurate with those used up to this point (FA rate, FID to seed, or “FID to perceptogram”). These results should be presented using measurements that can be directly compared with the previous data.

Thanks for the suggestion. We now report the image effects using the same measure (FID) as the rest of the paper.

As for the Miss rate, please note that FA and Miss are orthogonal measures, and here we need to use the Miss rate to support our claim. The claim here is that lowering the stimulation intensity takes a toll on the animals’ performance by decreasing the detectability of the perceptual event. In this case, we do not have any strong prediction about the FA rate, but we expect the Miss rate to increase.

3) Behavioral Measures:

The authors made good use of signal detection theory, but omitted a few important measures.

3a) Page 5, Figure 2: The interpretation of the shift in FA rate with exposure to the optimized perturbed images depends critically on those images remaining a constant fraction of all perturbed images. Was that the case? What was the overall fraction of the stimulus presentations involving the optimized stimulus versus all others?

No. The ratio of Ahab-optimized images decreased over time. But this cannot affect the probability of FA. Please note that the entire image set is balanced with respect to the stimulation condition. Every one of the Ahab-optimized image perturbations (including the perceptograms), as well as all DaVinci image perturbations, were shown an equal number of times with and without brain stimulation. The chance level for an image belonging to the

stimulation group was 50% for all image groups throughout the procedure. As a result, using 100% DaVinci or 100% Ahab or varying their ratios would not affect the change level for the task. So, decreasing the ratio of Ahab-optimized images would not result in having more FAs.

3b) Page 3: How was the miss rate kept fixed while increasing FA and not providing differential rewards for hits and correct rejections?

In our optimization, we aimed at evolving images that would increase the FA rate without increasing the Miss rate. Please note that if both FA and Miss rates increase, that would only mean that the task has become generally more difficult for the monkey in the presence of a given image. However, here we aim to find images that would be mistakenly taken as brain stimulation. Such an image would have the following effects on the signal detection theory measures: FA and Hit: increase, and Miss and CR: decrease. As shown in Figure 4.a., this is, in fact, the case. Perceptograms, as if they resemble the actual brain stimulation, increase the chances of reporting a trial as stimulated independent of the condition. Thus, the perceptograms: increase the chance of FA when there is no stimulation, increase the chance of Hit when there is stimulation (because they get aligned with the stimulation-induced effect and reinforce it), and decrease the Miss and CR rates. Note that this is a very similar effect compared to Newsome's classical findings and Afraz et al. 2006. In those studies, it has been shown that the visual stimulus which aligns with the perceptual outcome of brain stimulation changes the behavioral bias in favor of reporting stimulation (only for those stimuli); this increases both FA and Hits and decreases Miss and CRs. Here, we find basically the same effect (Fig 4.a.), but instead of reporting the criterion, in order to provide more clarity, we chose to show both Hits and FAs.

4) Unjustified claims:

4a) Page 3/Figure 1: The authors argue that the rapid rise in FAs when visual perturbations are introduced argues that the animals perceive the optogenetic stimulation as visual. The 50% FA rates suggest the animals initially generalized their cortical perturbation detection task to be a general perturbation detection task, such that all visual perturbations would be rewarded perturbation. After a few hundred trials, they had an "ah-ha" moment and stopped responding to the visual perturbations (which they presumably could distinguish throughout). Generalizing to other perturbations does not provide strong support for IT stimulation producing a visual experience.

We thank the reviewer for this pushback. The observation in Figure 1 shows that the animals can mistakenly take the visual perturbation as "cortical stimulation." The reviewer is correct that "cortical stimulation" being a visual event is not the only possible explanation for this. For instance, the monkey might experience an auditory effect following stimulation but generalize it to the visual perturbation before learning the task. We have softened the claim to reflect this point (page 3). Also, following the suggestion by Reviewer 1, we have updated Figure 1 and added more information to it.

4b) The authors claim that these images reveal and depict the contents of the complex hallucinatory percepts induced by neural perturbation in IT. That is an overstatement. They find that radically different images can substitute for optogenetic stimulation of a given cortical site (cf. the left 2 image columns or the right 4 image columns in Figure 5c). We can infer that each of these image groups captures some important component of the percept produced by IT stimulation, but none can be said to reveal or depict the contexts of that percept, or a “picture” of visual hallucination.

With permission from the reviewer, we defend our original claim here.

The fact that we find radically different perceptograms for different seed images does not mean that each explains only a small fraction of the variance of a much larger perceptual effect. The baseline FA rate is only ~4%, and perceptograms induce, on average, 70% FA with the theoretical maximum being 100%. This is a huge effect that explains the majority of the variance for each seed image. It is also very specific; the slightest change in the perceptogram makes the monkey notice it is not the actual stimulation event (note that the monkey FAs against reward). It is also -as shown- replicable and stable over time. Now, the fact that different seed images lead to different Perceptograms, each inducing ~70% FA rate, cannot be reconciled with the idea of “one” perceptual effect coming out of stimulation of a single point in IT cortex, independent of the image on the retina. We understand that the null hypothesis here is that stimulation of a given point in IT cortex induces a constant percept independent of what’s shown to the eyes. We find the current results exciting exactly because they go against this prior. Our earlier paper (Azadi et al., 2023, *Current Biology*) has already established this concept, and here we provide more evidence for the case.

4c) Page 1: The statement that animals “could not discriminate [the images] from the state of being cortical stimulated” needs to be qualified. Performance was with optogenetic stimulation was clearly distinguishable in the animals’ behavior.

The animals could not “discriminate” the state of being stimulated from the state of looking at perceptograms simply because they made the same behavioral response for both states in most of the trials (~70%). Of course, optogenetic stimulation could clearly be discriminated from no stimulation condition for most of the stimuli, but that only provides a baseline that strengthens the claim that it is hard for the animal to discriminate the stimulation state from looking at perceptograms.

4d) Page 1: The design of these experiments does not match standard definitions of “closed-loop” stimulation.

Thank you for bringing this possible confusion to our attention. We have now removed the term closed-loop from the manuscript.

4e) Page 1: The claims that these observations “enable” better visual prosthetic devices or relate directly to hallucinations in mental disorders also seem difficult to defend.

These findings help with the development of better visual prosthetics in two ways:

1. Systematic measurement of the perceptual effects of brain stimulation enables better use and calibration of visual prosthetic devices. However, accurate and unbiased measurement of the perceptual events induced by stimulation of the visual areas (even in the case of V1) is difficult. The methods presented in this study open the door to better measurement and more effective calibration of prosthetic devices.
2. Visual prosthetic devices traditionally target the primary visual cortex. This forces the system to recreate any visual scene with “phosphene” elements, the result of local stimulation in the primary visual cortex. However, consistent with the arguments of the following papers, it is not possible to recreate rich and complex visions only by phosphenes. The current manuscript documents the high-level visual effects induced by stimulation of IT cortex. This opens the door to adding high-level visual areas to the target areas of prosthetic devices.

As for the case of visual hallucinations, for instance, in Temporal Lobe Epilepsy (TLE), currently, there exists no mechanistic understanding of the possible underlying physiology. The findings of this paper at least constrain the space of possible mechanisms by documenting the shape and quality of the hallucinatory events induced by focal stimulation in IT cortex.

These points are now clarified in the paper to support the claims (page 11).

1 - Bosking, William H., Michael S. Beauchamp, and Daniel Yoshor. "Electrical stimulation of visual cortex: relevance for the development of visual cortical prosthetics." *Annual review of vision science* 3 (2017): 141-166.

2 - Lowery AJ, Rosenfeld JV, Lewis PM, Browne D, Mohan A, et al. 2015. Restoration of vision using wireless cortical implants: the Monash Vision Group project. *Conf. Proc. IEEE Eng. Med. Biol. Soc* 2015:1041–44

3 - Troyk P, Bak M, Berg J, Bradley D, Cogan S, et al. 2003. A Model for intracortical visual prosthesis research

4 - Luo YH, da Cruz L. 2016. The Argus R II Retinal Prosthesis System. *Prog. Retin. Eye Res* 50:89–107

5) Missing details:

5a) Was the optogenetic stimulation appropriately delayed for IT neuronal latencies?

This, in fact, is a very interesting question; when designing the study, we spent a lot of time deciding on this detail. While IT is known to have a response latency of ~70-80ms, it is not clear what part of the IT response causally contributes to perception. In that sense, any time

adjustment would be based on arbitrary assumptions about the decoding of the signal from the IT cortex. Thus in an ideal world, we need to measure the best time for stimulation systematically. At the beginning of the current study, which looked like a very risky project, we decided to limit our free parameters and leave this optimization for the follow-up studies. We argued that the potential subtle temporal difference in the timing of the events would be difficult for the monkeys to notice, especially in the stimulation-absent trials that are the source of the FAs. Moreover, if the animals noticed a potential time lag effect, they would use it in order to get more rewards and not be tricked by the perceptograms. Thus, we used a 150ms stimulation block, synchronized to the onset of the visual stimulus (no delay correction), which is long enough to cover the early IT response as well. We are currently doing another project using a much shorter stimulation train, specifically designed to find how the timing of stimulation interacts with perception. This information is now reflected in the manuscript (page 7).

5b) Page 15: "Lateral bank of the central IT cortex" does not narrow things down well enough for other to replicate these experiments.

Thanks for demanding this. We have now specified the anatomy. The array was positioned on the lateral convexity of TE cortex just below STS. The LED board spanned from 7mm to 12mm anterior to the interaural line, crossing from TEpd (dorsal posterior TE) to TEad (dorsal anterior TE) according to the Saleem and Logothetis atlas. This information is now reflected in the manuscript (pages 11-12).

5c) Page 15: Presumably the LED power was constant (not pulsed) during the 150 ms stimulation period, but this should be stated.

The LED power was constant during 150 ms of stimulation (square wave). This is now reflected in the paper (pages 11-12).

5d) Figure 6: The x-axes should be given in millimeters (or ranges of millimeters).

Fixed.

6) Page 15: A statement in the Methods suggests that the current results are based at least in part on data that have been previously published. There is no problem with reusing data, but the manuscript must include a statement making clear where the data have previously appeared, whether the current data overlap completely or in part, and whether any of the current results have been reported previously.

The results presented here are new and not reported earlier. The only place where we have used data from the earlier study is for a part of Figure 1.c., which shows the transition to the new paradigm during the training. Even this part of the data from our previous study was not published anywhere before and is unique to this manuscript.

7) Given that these two subjects were capable of detecting optogenetic stimulation in the absence of any visual stimulus, it would be helpful for the authors to discuss whether less biased images might be produced starting from a blank slate.

Thanks for demanding more discussion here. In our earlier studies (Azadi et al. 2023, Lafer-Sousa et al. 2023), we found that detecting cortical stimulation is difficult for the animals when looking at a blank screen. While the animals still perform the task above the chance level, introducing a visual stimulus dramatically increases their performance. Blending the visual stimulus into the background by reducing its contrast/visibility or its size takes a big toll from detection performance. This is consistent with a human study (Murphy et al. 2009) showing that humans barely notice stimulation of their fusiform cortex when blindfolded. This is also another line of evidence against the null prior discussed earlier, that brain stimulation induces a constant percept independent of the visual stimulus.

Given the lower dynamic range of performance for the blank screen as well as the technical complications of mutating blank images, we avoided perceptography with a blank seed for this initial study. We aim to explore this complicated matter systematically in our following studies.

These points are now reflected in the manuscript (page 3).

1- Azadi, R., Bohn, S., Lopez, E., Lafer-Sousa, R., Wang, K., Eldridge, M.A. and Afraz, A., 2023. Image-dependence of the detectability of optogenetic stimulation in macaque inferotemporal cortex. *Current Biology*.

2- Lafer-Sousa, R., Wang, K., Azadi, R., Lopez, E., Bohn, S. and Afraz, A., 2023. Behavioral detectability of optogenetic stimulation of inferior temporal cortex varies with the size of concurrently viewed objects—current Research in Neurobiology, 4, p.100063.

3- Murphey, D.K., Maunsell, J.H., Beauchamp, M.S. and Yoshor, D., 2009. Perceiving electrical stimulation of identified human visual areas. *Proceedings of the National Academy of Sciences*, 106(13), pp.5389-5393.

Minor Comments

8) I think it is up to the authors to decide whether they want to attach themselves to a newly forged label, however “perceptograms” seems a bit desperate given how these images likely to relate to percepts (Comment 1).

Given our earlier response to comment 4.b., we would prefer to stick to the term.

9) Page 15: “10 uLit” is not SI. The number of viral particles injected should be provided.

The numbers and units are now updated with respect to the SI system. Besides SI values, we have kept the uLit since it is a unit commonly used in optogenetics and helps those who want to replicate the procedures.

10) Pages 10-12: The material in “Perceptograms and natural image manifolds” would be more naturally presented in the Discussion.

Thanks for the suggestion. The entire section is now moved down and placed just before “Conclusion.”

11) Page 5: It is inappropriate to speak of “untraining” the monkeys in a signal detection theory task. What the authors probably mean is that the animals will adjust their behavioral criterion in response to sustained challenges from particular perturbed image sets.

We assume that the monkeys set an internal criterion to perform the task; for instance, this could make an internal criterion: “If that specific visual distortion happens, I should look up, and they will reward me.” Now, if we successfully mimic the visual distortion induced by stimulation, the monkey would follow the same internal rule and look up, but he won’t receive any reward for that. This will force the monkey to give up on his criterion for the task (the one we trained him to pay attention to) and change his internal strategy, looking for other clues that do not exist (task conditions are balanced and randomized, and there are no other clues to perform this task successfully)(please see Azadi et al., 2023, *Curr Bio*, for more details). As a result, the monkey will be “untrained” with respect to the task.

12) The y-axis in Figure 3a should start at zero. It is graphically misleading to do otherwise.

Thanks. Fixed.

13) The title is not informative about the experiments performed and seems to offer more than the data can provide.

Thanks. Changed.

14) Referring to the visual stimulus as a “video” encourages readers to incorrectly believe that the 1 s video was dynamic throughout. It would be safer to refer to it as 1 s sequence of static images.

Thanks. Fixed. And thank you for the helpfully detailed review.

REVIEWER COMMENTS

Reviewer #1 (Remarks to the Author):

The authors have greatly improved the manuscript, providing more clarity throughout, both in terms of data presentation and written text. The authors have also adequately addressed most of the reviewers' concerns. Along with another reviewer, I continue to question the authors' insistence on introducing the term "perceptogram" and linking the current study to mechanisms of "visual hallucinations". However, I defer to the journal editors on the suitability of the authors' choices.

Reviewer #2 (Remarks to the Author):

This already strong paper is improved. Issues from the first round linger, however.

My understanding of the FID is that is a function of two mean vectors and two covariance matrices. How is a covariance matrix obtained from the (presumed deterministic) features of a single image? How was the FID normalized? Simply know that the normalization was "based on the BigGAN interpolation factor" is not helpful to a non-specialist. The legend to figure 1d refers to the "[number of] non-class features included in each image produced by the engine (see Methods)" but the Methods cast little light on how this calculation is actually performed.

In several places in the text, the word "feature" is used in a technical way that some readers may not be familiar with (e.g. "... 82% (StD=21%) of the features in the perceptograms are shared with their corresponding seed images."). In other places, "features" appears to be more colloquial (e.g. "Neurons in the inferotemporal (IT) cortex respond selectively to complex visual features" and "... perceptograms of the anterior channels of the array tend to retain general features of the seed image..."). The paper could be improved by distinguishing these two uses of the word more clearly, and by defining the former use rigorously.

The responses to reviewers says that Supplementary Figure 2 includes the full collection of

the perceptograms and their seeds, but Supplementary Figure 2 is unchanged.

The ordinate of Figure 5b remains RMSE despite the response to Reviewer 3 that FID was used.

The ordinate of Figure 3b remains "FID to perceptogram" despite response to Reviewer 2 that the axis was labelled improperly.

Minor comments:

Page 2, typo: "the animal task remained the same"

Page 2, ambiguity: "...together with the Miss trials when a...". I think that the intended meaning is something like "...nor are the Miss trials, those on which a..." but I am not completely sure about this.

The title of Figure 1b is "The first training day with dynamic stimuli" but data from at least two days are shown.

"stimulation... distorts the perceive image by adding irrelevant visual features to the contents of perception". What does "relevance" mean in this context?

Page 11, typo: "phosphine" -> "phosphene"

Page 11: "This information is now reflected in the manuscript." This is sentence unnecessary.

Figure 6: "c" and "d" labels are missing.

Citation to reference #18 should be to reference #17 in "DaVinci, our illustrator engine, was built on BigGAN..."

Reviewer #3 (Remarks to the Author):

Shahbazi and his colleagues have revised their manuscript to address most of the concerns raised during the previous review.

Major Comments

2) Figure 5 has been improved by the addition of the images presented in the experiments, and an additional example. (Although the description “left SP, right Ph” in the legend is now ambiguous.) However, the authors have not addressed why measure the differences in this Figure 5 using RMSE, while FID-to-seed is used everywhere else in the manuscript. These data should be expressed in using FID so the differences between high and low stimulus conditions can be directly compare with other differences presented.

4c) The statement that animals “could not discriminate [the images] from the state of being cortical stimulated” needs to be qualified. Performance was with optogenetic stimulation was clearly distinguishable in the animals’ behavior.

Authors response: "The animals could not “discriminate” the state of being stimulated from the state of looking at perceptograms simply because they made the same behavioral response for both states in most of the trials (~70%). Of course, optogenetic stimulation could clearly be discriminated from no stimulation condition for most of the stimuli, but that only provides a baseline that strengthens the claim that it is hard for the animal to discriminate the stimulation state from looking at perceptograms."

If the animals could not discriminate “perceptograms” from optogenetic stimulation, they would give the same behavioral response across all those trials. Saying the discrimination is hard is not the same as saying the animals “did not discriminate [the images] from the state of being cortically stimulated” or that the image perturbations were “exchangeable with the state of being cortically stimulated”.

Reviewer #1 (Remarks to the Author):

The authors have greatly improved the manuscript, providing more clarity throughout, both in terms of data presentation and written text. The authors have also adequately addressed most of the reviewers' concerns. Along with another reviewer, I continue to question the authors' insistence on introducing the term "perceptogram" and linking the current study to mechanisms of "visual hallucinations." However, I defer to the journal editors on the suitability of the authors' choices.

We thank the reviewer for finding most of our responses adequate.

As for the term "perceptogram" and the link to "visual hallucinations," with all respect, we would prefer to stand by our wording choices. The majority of "decoding" studies of the visual system are concerned with finding the "best stimuli" for visual neurons (e.g., Ponce et al., Bashivan et al.) or reconstruction of the external visual stimulus that creates a given brain state (e.g., Ozcelik et al., Du et al.). However, our study fundamentally differs from this background in that we "decode" not the external stimulus, but the internal perceptual state induced by brain stimulation. In this sense, our results are directly linked to visual hallucinations, and the term "perceptogram" helps clarify this important point of departure from the earlier studies.

Ozcelik, Furkan, Bhavin Choksi, Milad Mozafari, Leila Reddy, and Rufin VanRullen.
"Reconstruction of perceived images from fMRI patterns and semantic brain exploration using instance-conditioned gans." In *2022 International Joint Conference on Neural Networks (IJCNN)*, pp. 1-8. IEEE, 2022.

Du, C., Li, J., Huang, L. and He, H., 2019. Brain encoding and decoding in fMRI with bidirectional deep generative models. *Engineering*, 5(5), pp.948-953.

Reviewer #2 (Remarks to the Author):

This already strong paper is improved. Issues from the first round linger, however. My understanding of the FID is that it is a function of two mean vectors and two covariance matrices. How is a covariance matrix obtained from the (presumed deterministic) features of a single image? How was the FID normalized? Simply knowing that the normalization was "based on the BigGAN interpolation factor" is not helpful to a non-specialist. The legend to figure 1d refers to the "[number of] non-class features included in each image produced by the engine (see Methods)" but the Methods cast little light on how this calculation is actually performed.

We thank the reviewer for the encouraging words. As for the questions regarding using FID, please find below our point-to-point responses.

How is a covariance matrix obtained from the (presumed deterministic) features of a single image?

FID, when measured for a single image, has a variance of zero and represents only the distance between average feature matrices.

We use the opportunity here to explain an important nuance in the measure of feature distance utilized in our study. While traditional FID measures use only the dissimilarity matrix from the last layer (Pool 3) of a feedforward neural network, when a measure of feature distance is intended to be used for a single image (compared to a reference one), it is better to compare the dissimilarity matrices from multiple layers of the underlying neural network. This modified FID measure produces a dissimilarity index that is well correlated with human perceptual dissimilarity judgments (Dosovitskiy et al). Thus, in the absence of covariance, this modified FID measure is technically called "perceptual loss". Since the term "perceptual loss" can be very confusing in the context of our paper (where the animal's perception is the main point of interest), we just used the general term "FID distance" to refer to our measure of feature distance.

In light of the reviewer's legitimate demand for more clarity here, we have now explained this point clearly in the methods section (page 13).

Dosovitskiy, Alexey, and Thomas Brox. "Generating images with perceptual similarity metrics based on deep networks." *Advances in neural information processing systems* 29 (2016).

How was the FID normalized? Simply know that the normalization was "based on the BigGAN interpolation factor" is not helpful to a non-specialist. The legend to figure 1d refers to the "[number of] non-class features included in each image produced by the engine (see Methods)" but the Methods cast little light on how this calculation is actually performed.

Thank you for asking for more clarity here. Firstly, please note that the distance measure in Figure 1d is not FID. In this case, we were interested to see if the addition of features to a certain image class would affect the FA profile or not. Thus, we simply used the BigGAN interpolation factor, the number of non-class features that were mixed into the seed by the generator engine itself. This measure -if the reviewer demands- can be turned into FID distance, but that would require FID normalization across different classes, which we avoided wherever possible as we preferred less processed measures. In either case, the point of this plot is that more image changes do not necessarily induce more FAs. This point has also been shown by an RMSE measure (added in the past review).

In a few cases, specifically Figures 3. b, 5. c, and 6.d, we needed a comparable measure of the distance across multiple perceptogram families; thus, the FID measure had to be normalized. In those cases, we normalized the FID values from each image pool to the maximum FID observed in that pool.

We have clarified these points and definitions in the manuscript (pages 13, also see Fig 1.d caption).

In several places in the text, the word "feature" is used in a technical way that some readers may not be familiar with (e.g. "... 82% (StD=21%) of the features in the perceptograms are shared with their corresponding seed images."). In other places, "features" appears to be more colloquial (e.g. "Neurons in the inferotemporal (IT) cortex respond selectively to complex visual features" and "... perceptograms of the anterior channels of the array tend to retain general features of the seed image..."). The paper could be improved by distinguishing these two uses of the word more clearly, and by defining the former use rigorously.

The reviewer is right. We have now revised the text and used "feature vector" wherever a mathematical definition of "feature" is intended. In the case of Yolo results, we dropped the usage of "feature" altogether and used the technical term "class label".

The responses to reviewers says that Supplementary Figure 2 includes the full collection of the perceptograms and their seeds, but Supplementary Figure 2 is unchanged.

You are right, we should have been more clear in our response letter. Supplementary Figure 2 includes the full collection of "high-intensity" perceptograms. We did not include the low-intensity perceptograms here because the changes were subtle and difficult to notice in the image

thumbnails used for the supplementary figure. The figure legend is now modified to avoid ambiguity. A few examples of the low-intensity perceptograms are already presented in Figure 5 in larger sizes.

The ordinate of Figure 5b remains RMSE despite the response to Reviewer 3 that FID was used.

Thanks for noticing. Fig 5. b is intended only to quantify the changes in the heatmaps with the most direct and least complicated measure (RMSE). While keeping the RMSE (for the sake of simplicity), we have now added the same result in FID units as well (Fig 5. c).

The ordinate of Figure 3b remains "FID to perceptogram" despite response to Reviewer 2 that the axis was labelled improperly.

We apologize for the error and thank the reviewer for noticing it. It is fixed now.

Minor comments:

Page 2, typo: "the animal task remained the same"

Fixed.

Page 2, ambiguity: "...together with the Miss trials when a...". I think that the intended meaning is something like "...nor are the Miss trials, those on which a..." but I am not completely sure about this.

The reviewer's interpretation is correct. The ambiguity is clarified now.

The title of Figure 1b is "The first training day with dynamic stimuli" but data from at least two days are shown.

Thanks, fixed.

"stimulation... distorts the perceive image by adding irrelevant visual features to the contents of perception". What does "relevance" mean in this context?

The word "irrelevant" is swapped with "unrelated" for more clarity.

Page 11, typo: "phosphine" -> "phosphene"

Thanks, fixed.

Page 11: "This information is now reflected in the manuscript." This is sentence unnecessary.

Oops! Sorry about that, fixed.

Figure 6: "c" and "d" labels are missing.

Thanks, added.

Citation to reference #18 should be to reference #17 in "DaVinci, our illustrator engine, was built on BigGAN..."

Thanks, fixed.

Reviewer #3 (Remarks to the Author):

Shahbazi and his colleagues have revised their manuscript to address most of the concerns raised during the previous review.

We are happy that the reviewer has found most of our responses satisfying. Following, we try to address the remaining issues point by point.

Major Comments

2) Figure 5 has been improved by the addition of the images presented in the experiments, and an additional example. (Although the description “left SP, right Ph” in the legend is now ambiguous.) However, the authors have not addressed why measure the differences in this Figure 5 using RMSE, while FID-to-seed is used everywhere else in the manuscript. These data should be expressed in using FID so the differences between high and low stimulus conditions can be directly compare with other differences presented.

Thanks for demanding more clarity here, and sorry for the miscommunication in our previous response. Please note Fig 5. b is intended only to quantify the changes in the heatmaps. For this purpose, RMSE provides a more direct and less processed measure. While keeping the RMSE, we have now added the same result in FID units as well (Fig 5. c).

4c) The statement that animals “could not discriminate [the images] from the state of being cortically stimulated” needs to be qualified. Performance was with optogenetic stimulation was clearly distinguishable in the animals’ behavior.

Authors response: "The animals could not “discriminate” the state of being stimulated from the state of looking at perceptograms simply because they made the same behavioral response for both states in most of the trials (~70%). Of course, optogenetic stimulation could clearly be discriminated from no stimulation condition for most of the stimuli, but that only provides a baseline that strengthens the claim that it is hard for the animal to discriminate the stimulation state from looking at perceptograms."

If the animals could not discriminate “perceptograms” from optogenetic stimulation, they would give the same behavioral response across all those trials. Saying the discrimination is hard is not the same as saying the animals “did not discriminate [the images] from the state of being cortically stimulated” or that the image perturbations were “exchangeable with the state of being cortically stimulated”.

The reviewer is absolutely right. We have softened the language following the reviewer’s suggestion.

REVIEWERS' COMMENTS

Reviewer #1 (Remarks to the Author):

The revised manuscript is much improved - and I have no further comments to add.

Reviewer #2 (Remarks to the Author):

"This measure uses the feature-vector from multiple layers..." Were the default values from the <https://github.com/mseitzer/pytorch-fid> package used? If not, which layers were used in this calculation?

"...is shown to strongly correlate with human visual quality judgements (ref)". A citation (to Dosovitskiy et al. 2016?) should be included.

In the DaVinci section of the Methods: "mean pixel similarity = 17.44%". Should "similarity" here be "distance"?

I urge the authors to at least briefly explain what the "BigGAN interpolation factor" is (as they sort of did in their responses to the reviewers). A Google search on the quoted strong "BigGAN interpolation factor" yielded no hits.

Reviewer #2 (Remarks to the Author):

"This measure uses the feature-vector from multiple layers..." Were the default values from the <https://github.com/mseitzer/pytorch-fid> package used? If not, which layers were used in this calculation?

We calculated all four options, and for the charts, we used the 2048 layer.

"...is shown to strongly correlate with human visual quality judgments (ref)". A citation (to Dosovitskiy et al. 2016?) should be included.

Thanks, Fixed.

In the DaVinci section of the Methods: "mean pixel similarity = 17.44%". Should "similarity" here be "distance"?

Thanks, Fixed.

I urge the authors to at least briefly explain what the "BigGAN interpolation factor" is (as they sort of did in their responses to the reviewers). A Google search on the quoted strong "BigGAN interpolation factor" yielded no hits.

Thanks for asking for more clarity, but please note that Fig 1. d's captions already include an explanation of the interpolation factor, as we mentioned in our previous response.